# MorphoDiff: Cellular Morphology Painting with Diffusion Models

**Zeinab Navidi**[1,2,3]**, Jun Ma**[3,9]**, Esteban A. Miglietta**[4]**, Le Liu**[4]**, Anne E. Carpenter**[4]**, Beth A. Cimini**[4]**,
Benjamin Haibe-Kains**[3,5,6,7*†]**, Bo Wang** [1,2,3,5,8,9*†]

1. Department of Computer Science, University of Toronto, Toronto, ON, Canada
2. Peter Munk Cardiac Centre, University Health Network, Toronto, ON, Canada
3. Vector Institute, Toronto, ON, Canada
4. Imaging Platform, Broad Institute of Harvard and MIT, Cambridge, MA, USA
5. Department of Medical Biophysics, University of Toronto, Toronto, ON, Canada
6. Structural Genomics Consortium, Toronto, ON, Canada
7. Princess Margaret Cancer Centre, University Health Network, Toronto, ON, Canada
8. Department of Laboratory Medicine and Pathobiology, University of Toronto, Toronto, ON, Canada
9. AI Hub, University Health Network, Toronto, ON, Canada
∗ Corresponding authors, † Supervised this work equally
`benjamin.haibe.kains@utoronto.ca, bowang@vectorinstitute.ai`

## Abstract

Understanding cellular responses to external stimuli is critical for parsing biological mechanisms and advancing therapeutic development. High-content image-based assays provide a cost-effective approach to examine cellular phenotypes induced by diverse interventions, which offers valuable insights into biological processes and cellular states. We introduce MorphoDiff, a generative pipeline to predict high-resolution cell morphological responses under different conditions based on perturbation encoding. To the best of our knowledge, MorphoDiff is the first framework capable of producing guided, high-resolution predictions of cell morphology that generalize across both chemical and genetic interventions. The model integrates perturbation embeddings as guiding signals within a 2D latent diffusion model. The comprehensive computational, biological, and visual validations across three open-source Cell Painting datasets show that MorphoDiff can generate high-fidelity images and produce meaningful biology signals under various interventions. We envision the model will facilitate efficient in silico exploration of perturbational landscapes towards more effective drug discovery studies.

## 1 Introduction

Recent advancements in generative artificial intelligence (AI) have propelled significant progress across various domains, including computer vision, natural language processing, and healthcare (Rombach et al., 2022; OpenAI, 2022; Meskó & Topol, 2023). Notable advancements in AI methodologies, coupled with the availability of vast datasets, have yielded foundation models with extensive applications and capabilities. Their practical application holds great promise for empowering practitioners in more effective drug development, thereby streamlining this process at scale while conserving expert and financial resources (Moor et al., 2023).

Modeling cellular responses to external interventions is a key focus in computational biology; it aims to uncover biological insights that can inform the design of more effective therapies, which are, in essence, interventions aiming at a particular cellular response (Celik et al., 2024). While there have been significant efforts to model cellular dynamics using AI algorithms, much of the prior work has concentrated on transcriptomic-level changes (Roohani et al., 2023; Lotfollahi et al., 2023; 2019; Cui et al., 2024; Hetzel et al., 2022). However, advances in high-throughput screening technologies now enable the exploration of rich phenotypic readouts, such as those generated by high-content microscopy imaging, which provide critical insights into cellular activity and accelerate drug target identification and mode-of-action studies under diverse conditions (Seal et al., 2024). Among these technologies, the Cell Painting assay—a high-content microscopy imaging platform—has emerged

as a powerful, cost-effective approach for cellular phenotype screening, playing a pivotal role in understanding the morphological characteristics of cells under various perturbations (Bray et al., 2016). Image analysis software, such as CellProfiler, has been widely adopted to extract detailed features from microscopy images (Carpenter et al., 2006; Chow et al., 2022; Moshkov et al., 2024). These features have provided valuable insights into compound polypharmacology (Chow et al., 2022), mechanisms of action (Tian et al., 2023; Way et al., 2022; Wong et al., 2023; Dee et al., 2024), and target genes associated with specific perturbations (Way et al., 2022). Despite these advances, a significant challenge in virtual screening remains: the size of existing screening libraries represents only a small fraction of the vast chemical space, which is estimated to contain over $10^{60}$ drug-like molecules (Lu et al., 2024; Lipinski et al., 2012; Reymond, 2015). This limitation is particularly critical when seeking to identify the most effective treatments for specific cellular conditions. Machine learning (ML) and generative models might overcome this barrier by enabling response estimation across a much larger perturbational space, thus improving treatment efficacy while substantially reducing cost.

There is growing interest in leveraging advanced deep learning methods to directly learn cellular patterns from high-content images, rather than relying solely on engineered features extracted using traditional image analysis software. A notable example is a recent study that developed a strategy for learning representations of treatment effects from high-throughput imaging using a causal framework (Moshkov et al., 2024). By employing weakly supervised learning, the model captured both confounding factors and phenotypic features in the learned representations, providing a comprehensive view of treatment-induced changes. Another significant contribution is a retrieval system based on multi-modal contrastive learning, which maps molecular perturbation and their corresponding Cell Painting image features into a unified embedding space (Sanchez-Fernandez et al., 2023). Although the model lacks generative capabilities, it proves to be a valuable tool for retrieving perturbations with morphological effects most similar to those of the input query. Two relevant studies used generative models to transfer cell style between conditions in low-resolution, single cell cropped image patches (Palma et al., 2023; Bourou et al., 2023). Although these methods offer insights into cellular transitions, their focus on isolated cells overlooks inter-cellular effects in a wide well area as screened in original microscopy images and limits practical applicability. The most relevant method is a pipeline that employs conditional flow models for cellular phenotype estimation using perturbation information that was only tested on molecular interventions (Yang et al., 2021).

To the best of our knowledge, this work represents one of the first efforts for estimating cell responses at a high-resolution and high-content image scale. By leveraging advanced generative models in computer vision, as exemplified by Rombach et al. (2022), and incorporating state-of-the-art (SOTA) perturbation encoding modules, including the single-cell foundational model proposed by Cui et al. (2024), we introduce MorphoDiff, a novel diffusion-based generative pipeline for high-resolution cellular morphology prediction, generalizable to both genetic and chemical perturbations. We benchmarked MorphoDiff on three microscopy imaging datasets, evaluating image fidelity and biological properties of cells in in-distribution and unseen perturbations, demonstrating promising performance overall. This work showcases the potential of advanced generative models to predict high-resolution cellular phenotypic responses across a broad range of perturbations, and we hope it paves the way for future research in this direction.

## 2 METHOD

Diffusion models have emerged as a powerful approach for image generation, overcoming challenges such as training instability and mode collapse common in GAN-based architectures. In this work, we developed MorphoDiff, a novel pipeline capable of generating perturbed cellular morphology in high-resolution and high-content microscopy images with the integration of perturbation embedding as a guiding signal within the generative model. Our pipeline contains two key components: an image-based denoising diffusion probabilistic model (DDPM) and a perturbation projection module. The schematic representation of MorphoDiff is illustrated in Figure 1.

### 2.1 LATENT DIFFUSION MODEL FOR IMAGE GENERATION

The Stable Diffusion (SD) framework is built upon a latent diffusion model (LDM) pre-trained on a vast corpus of natural images, enabling the generation of realistic images based on input text prompts

(Rombach et al., 2022). We adapted SD pipeline for our task, and transformed available Cell Painting channels (elaborated in the 3.1 section) into three RGB channels for modelling. For our specific task, we fine-tuned the pre-trained DDPM model in the SD framework on Cell Painting images projected into latent space. We opted to employ the existing SD VAE trained on natural images due to its demonstrated efficacy in reconstructing Cell Painting images (sample images provided in Appendix Figure 5). More specifically, given a high resolution $512 \times 512$ RGB image $X$, the image encoder $E$ transforms $X$ into a latent representation $z_0 = E(X)$ consisting of four channels of $64 \times 64$ feature maps. Subsequently, the decoder $D$ reconstructs the generated image in latent space as the final outcome. The DDPM model learns sample distribution in the latent space, and perturbs the latent image representation $z$ by introducing noise during the forward diffusion process.

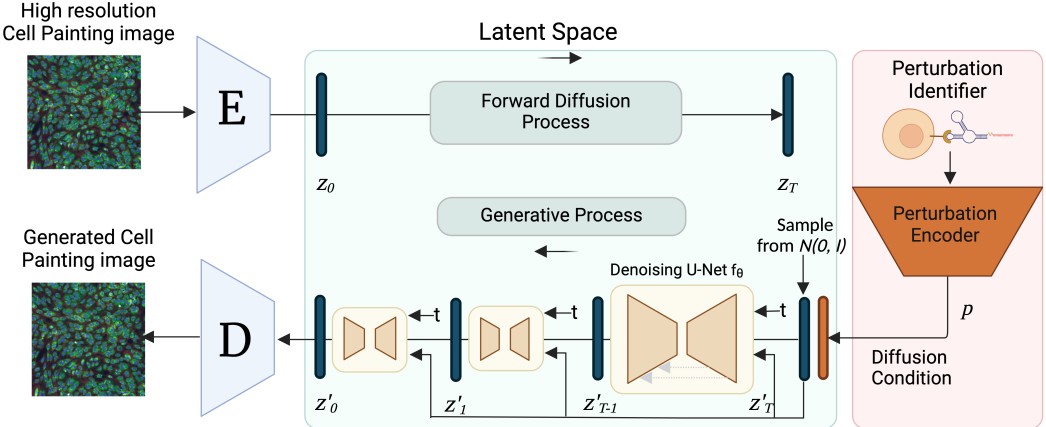

Figure 1: **MorphoDiff overview:** Workflow diagram illustrating the architecture of the MorphoDiff pipeline, comprising two primary modules: a DDPM module and a perturbation encoder. The diffusion algorithm serves as the central component of the workflow, performing conditional image generation. The perturbation projection module can be adapted depending on perturbation type.

In particular, the LDM training consists of two steps: a forward diffusion process and a generative (or sampling) process. The forward process is conducted on the latent variable $z$, generating a noisy perturbed embedding at various time points $t$ from $0$ to $T$. This process is done by adding Gaussian noise at each step, which is defined by:

$$q(z_t|z_{t-1}) = \mathcal{N}(z_t; \sqrt{1-\beta_t}z_{t-1}, \beta_t I), \tag{1}$$

where $\beta = \beta_1, \cdots, \beta_T$ is the pre-defined variance schedule. It has been shown that the distribution of noisy samples converges to a standard Gaussian distribution as $t$ approaches $T$ (Ho et al., 2020). Assuming that the noise values follow a Gaussian distribution, $z_t$ can be sampled based on $z_{t-1}$ at any desired time step $t$ in a closed form using the reparameterization trick:

$$z_t = \sqrt{\alpha_t}z_{t-1} + \sqrt{1-\alpha_t}\epsilon_{t-1}, \tag{2}$$

where $\alpha_t = 1 - \beta_t$ and $\epsilon \sim \mathcal{N}(0, I)$. By using the chain rule and recalling that merging multiple Gaussian distributions remains Gaussian, we can derive a direct formula for $z_t$ from $z_0$ as follows

$$z_t = \sqrt{\bar{\alpha}_t}z_0 + \sqrt{1-\bar{\alpha}_t}\epsilon, \tag{3}$$

where $\bar{\alpha}_t = \prod_1^t \alpha_i$, and $\epsilon$ represents a sample noise drawn from a standard Gaussian distribution.

The forward process generates the noisy perturbed image embedding $z_t$ at time step $t$. A 2D U-Net is trained to estimate the noise level from the noisy embedding $z_t$ (Ronneberger et al., 2015). In this process, the projected perturbation embedding $p$ is integrated as a condition along with the $z_t$ and time variable $t$, and are fed into the U-Net to predict the noise. The conditional LDM algorithm in MorphoDiff leverages the information from $p$ to guide the conditional denoising step and perturbation-specific phenotype estimation.

The MorphoDiff pipeline optimizes the U-Net model prediction by minimizing the Mean Squared Error (MSE) between the ground truth noise $\epsilon$ and the predicted noise. The loss enables the network

to learn and accurately predict the added noise for each perturbed sample at time point $t$, facilitating the denoising process. The objective of the diffusion model can be summarized as:

$$loss = MSE(f_\theta(p, \sqrt{\bar{\alpha}_t}z_0 + \sqrt{1 - \bar{\alpha}_t}\epsilon, t), \epsilon), \tag{4}$$

where $f_\theta$ denotes the U-Net model. During the sampling process, MorphoDiff generates images based on the input perturbation. To initiate the generation process, a random noise vector is then sampled from a standard Gaussian distribution, denoted as $z'_T$. The pipeline iteratively calculates $z'_t$ for $t$ ranging from $T$ to 0 by:

$$z'_{t-1} = \frac{1}{\sqrt{\alpha_t}} \left( z'_t - \frac{1 - \alpha_t}{\sqrt{1 - \bar{\alpha}_t}} f_\theta(p, z'_t, t) \right) + \sqrt{\beta_t}\epsilon, \tag{5}$$

where $z'_{t-1}$ represents a less noisy embedding of the generated perturbed image at time step $t - 1$. It is obtained using the current embedding $z'_t$, the prediction $f_\theta(p, z'_t, t)$ made by the U-Net model, and a random noise sample $\epsilon$. The coefficients $\alpha_t$ and $\beta_t$ control the contributions of the embedding and the noise, respectively. Through this iterative process, MorphoDiff generates a contextually relevant embedding that captures the desired cellular phenotype.

## 2.2 PERTURBATION PROJECTION MODULE

The original SD pipeline employed the CLIP model as the text encoder (Radford et al., 2021; Rombach et al., 2022). To generate images exhibiting cellular morphological changes under specific treatments, we replaced CLIP with the perturbation embeddings obtained from projection algorithms depending on the type of perturbations. Our pipeline provides the flexibility of integrating different projection modules and can handle both genetic and chemical interventions.

For genetic perturbations, we used the SOTA single-cell foundation model scGPT, which has been trained on 33 million normal human cells and employs stacked transformer layers to generate cell and gene embedding simultaneously (Cui et al., 2024). scGPT has demonstrated strong ability to implicitly encode gene relationships in gene embeddings through generative modeling of gene expression (mRNA profiles). This property is particularly relevant for our task as it facilitates learning meaningful patterns of cellular morphology linked to the encoded gene representation containing transcriptomic signals, thereby enhancing generalizability to unseen perturbations.

For chemical compounds, we employed the molecular encoder software RDKit, a well-established tool that converts standard molecular representations in SMILES format into numerical embedding (Landrum, 2023). It therefore captures structural similarity of the chemical compounds. The projected latent variables are then fed into the MorphoDiff pipeline as a guiding signal. Further details on the perturbation incorporation steps (Appendix Note A.3.1), training protocol (Appendix Note A.3.2), and experiment information (Appendix Table 4) are provided in the Appendix section.

# 3 EXPERIMENTS

## 3.1 DATASET

Three publicly available Cell Painting datasets were used for modelling and validation, representing a diverse range of perturbations and cell types, described as follows.

**RxRx1 dataset** contains 1108 perturbations across four cell types (Sypetkowski et al., 2023). Each sample comprises six-channel fluorescent microscopy images capturing key cellular structures. Our analysis concentrated on the largest group within the dataset (HUVEC cell line) with samples drawn from multiple experimental batches. Each small interfering RNA perturbation in the dataset targets a gene, resulting in significant mRNA knockdown and corresponding changes in protein expression (Sypetkowski et al., 2023). We conducted two sets of experiments: one using all HUVEC images (All Batches), and the other constrained to a single batch (Single Batch) to minimize batch effects. The authors' provided code was used to convert the six-channel images into RGB format [1].

**BBBC021 dataset** comprises 13200 images of MCF7 breast cancer cells, stained for DNA, F-actin, and B-tubulin, and imaged using three-channel fluorescent microscopy, which were directly mapped

---

[1] Link to RxRx1 conversion code

to RGB format for modelling (Caie et al., 2010). The MCF7 cells were treated with 113 small molecules, each administered at eight different concentrations. We conducted two sets of experiments: one using all compounds for which SMILES-based projections were available and the embedding was generated, and another included 14 compounds obtained from a list of six mechanisms of actions (MOA) reported having distinct phenotypes (Caie et al., 2010).

**Rohban et al. dataset** contains U2OS cell images with 323 over-expressed genes (Rohban et al., 2017). Based on expert consultation, three of the five imaging channels—RNA, Mitochondria, and DNA— were selected for modeling and validation, prioritizing channels essential for image segmentation and feature extraction that are also biologically informative and interpretable (Stirling et al., 2021; Carpenter et al., 2006). Two gene subsets were analyzed, including: (1) five genes from pathways reported to affect cellular morphology, and (2) a list of 12 genes obtained from clustering performed in Rohban et al. (2017) based on gene morphological features. Detailed description of perturbations for each dataset and their download links are provided in Appendix Note A.3.3.

**Pre-processing:** All images with larger than $512 \times 512$ pixels were resized to $512 \times 512$ resolutions to ensure consistent dimensions, with the largest possible area covered in each image. Normalization and scaling followed best practices in image pre-processing (Rombach et al., 2022). More explanation of pre-processing are provided in Appendix Note A.3.4.

## 3.2 COMPARISON

For benchmarking, a set of comparative analysis was conducted against the fine-tuned unconditional SD as the baseline (with fixed prompt encoding), assessing the impact of integrating perturbation data on guiding the generative process to mimic cellular phenotype consistent with real signals. To the best of our knowledge, only one existing work offers conditional generative capabilities in a high-resolution setting comparable to our approach (Yang et al., 2021). However, due to technical challenges, adapting their model for benchmarking on our datasets was not feasible. To provide an assessment of MorphoDiff's performance against the SOTA, we benchmarked against an exiting method that generates cellular phenotypes in individual cell cropped patches (Palma et al., 2023).

For experimental validation, 500 images were generated per perturbation. Real (ground truth) images were augmented to match this number where necessary, using random flipping and rotation, to ensure consistency in evaluation metrics. The two-sample t-test was conducted for statistical analysis, with a p-value threshold of $< 0.05$ considered as statistically significant. Further evaluation details are provided in Appendix Note A.3.5.

## 3.3 RESULTS AND DISCUSSION

In this section, we present the results of MorphoDiff in predicting perturbation-specific cellular morphology, including image fidelity metrics, visual assessment, and biologically interpretable features.

### 3.3.1 MORPHODIFF IMPROVES PERTURBATION-SPECIFIC DISTRIBUTIONAL DISTANCE

To assess MorphoDiff's effectiveness to improve similarity of pixel distributions in generated images to the associated real images for each perturbation, we employed well-established metrics for validating the quality of images created by generative models, including Fréchet Inception Distance (FID) and Kernel Inception Distance (KID) (Heusel et al., 2017; Sutherland et al., 2018). FID summarizes the distance between the Inception feature vectors for real and generated images in the same domain, while KID measures the dissimilarity between two probability distributions using independently drawn samples. We computed these distance metrics between generated images for each perturbation and their corresponding real image groups.

Table 1 presents the average of calculated FID and KID distances across all perturbations. Lower FID and KID values indicate better alignment between generated and real distributions, with statistically significant differences between MorphoDiff and the unconditional Stable Diffusion highlighted in bold. Our experiments showed that MorphoDiff consistently outperformed the baseline across all experiments, by effectively embedding perturbation-specific patterns for both genetic and chemical interventions. In the RxRx1 (All Batches) and BBBC021 (All Compounds) experiments, distance metrics improved compared to smaller, single-batch experiments, suggesting that larger training sets enhance model performance. Additionally, we observed that MorphoDiff predictions improved

the average ranking of images to the matched real perturbed cohort based on the distance metrics. Further details are provided in Appendix Table 5.

Table 1: Average FID ($\times 10^{-2}$) and KID metrics across all perturbations in each experiment, assessing distributional similarity between generated and real images from the corresponding perturbation conditions (lower is better). Bold values highlight statistically significant differences with p-value $< 0.05$, * indicating p-value $< 0.01$ and ** indicating p-value $< 0.001$.

| Dataset | Experiment | Method | FID↓ | KID↓ |
|---|---|---|---|---|
| RxRx1 | All Batches | MorphoDiff | **0.78**** | **0.05**** |
| RxRx1 | All Batches | Stable Diffusion | 1.15 | 0.11 |
| RxRx1 | Single Batch | MorphoDiff | **1.14**** | **0.12*** |
| RxRx1 | Single Batch | Stable Diffusion | 1.45 | 0.16 |
| BBBC021 | All Compounds | MorphoDiff | **1.99**** | **0.21**** |
| BBBC021 | All Compounds | Stable Diffusion | 3.84 | 0.47 |
| BBBC021 | 14 Compounds | MorphoDiff | **2.26*** | **0.30** |
| BBBC021 | 14 Compounds | Stable Diffusion | 3.22 | 0.42 |
| Rohban et al. | 5 Genes | MorphoDiff | **2.51*** | **0.33*** |
| Rohban et al. | 5 Genes | Stable Diffusion | 3.26 | 0.45 |
| Rohban et al. | 12 Genes | MorphoDiff | **2.77*** | **0.38**** |
| Rohban et al. | 12 Genes | Stable Diffusion | 3.17 | 0.45 |

We further benchmarked MorphoDiff-generated images at the single-cell cropped patch scale against another existing method by Palma et al. (2023), with results summarized in Table 2 and details of the analysis provided in Appendix Note A.3.5. Statistical testing demonstrated that MorphoDiff outperforms the second-best method across all compounds except Cytochalasin B. For this compound, our investigation revealed that in larger (broader field of view) images, MorphoDiff achieved superior FID (1.9 vs. 2.9 ($\times 10^2$)) and KID (0.23 vs. 0.37) metrics compared to Stable Diffusion, aligning with our qualitative assessment (sample images provided in Appendix Figure 6). We propose that generating and validating images with larger fields of view facilitates capturing cellular density and intercellular relationships. This approach has the potential to provide holistic insights into cellular interactions, phenotypic shifts induced by various perturbations, and cellular diversity within a well.

Table 2: Average FID ($\times 10^{-2}$) and KID metrics assessing distributional similarity between generated and real images of cell cropped patches (lower is better). Bold values highlight statistically significant differences of the best model with the second best model with p-value $< 0.05$, * indicating p-value $< 0.01$ and ** indicating p-value $< 0.001$.

| Model | AZ138 | | AZ258 | | Taxol | | Cytochalasin B | | Vincristine | |
|---|---|---|---|---|---|---|---|---|---|---|
| | FID↓ | KID↓ | FID↓ | KID↓ | FID↓ | KID↓ | FID↓ | KID↓ | FID↓ | KID↓ |
| Stable Diffusion | 1.40 | 0.14 | 0.94 | 0.08 | 1.50 | 0.15 | **0.83**** | **0.07**** | 1.83 | 0.20 |
| IMPA | 0.98 | 0.07 | 1.18 | 0.12 | 1.28 | 0.11 | 1.23 | 0.11 | 1.05 | 0.07 |
| MorphoDiff | **0.82**** | **0.06**** | **0.76**** | **0.05**** | **1.09**** | **0.10** | 1.19 | 0.11 | **0.86**** | **0.06**** |

### 3.3.2 MORPHODIFF CAPTURES PERTURBATION-SPECIFIC CELL MORPHOLOGY SIGNALS

Among the three datasets, BBBC021 stands out due to its strong, visually detectable phenotypic changes induced by small molecule treatments, as well as for including several compounds with annotated Mechanisms of Action (MOA) (Caie et al., 2010). This dataset is particularly valuable for modeling and validating the biological interpretability of generated images, motivating us to examine MorphoDiff's ability to capture the biological signals associated with different conditions. Our visual assessment reveals that MorphoDiff effectively captures perturbation induced cellular patterns for distinct compounds, closely mimicking the morphology of treated cells in real images. In contrast, the baseline SD model primarily learns general cellular structures. Figure 2 presents sample images from six sample compounds representing the different MOA groups in the BBBC021 experiment (14 compounds). Our investigation further revealed that MorphoDiff-generated images successfully captured relatively rare cell cycle events, with sample images provided in Appendix

Figure 7. Overall, the qualitative comparison between real images and those generated by MorphoDiff highlights MorphoDiff's promising alignment with real phenotypes.

Additional analysis was performed by training and validating models on images obtained by cropping $512 \times 512$ pixel patches from the original BBBC021 images to assess the impact of focusing on a smaller well area. Further description of pre-processing and sample images for all experiments are provided in Appendix Note A.3.4, Appendix Figure 8 and 9, with image distance validation results of the cropped version of experiments provided in Appendix Table 6.

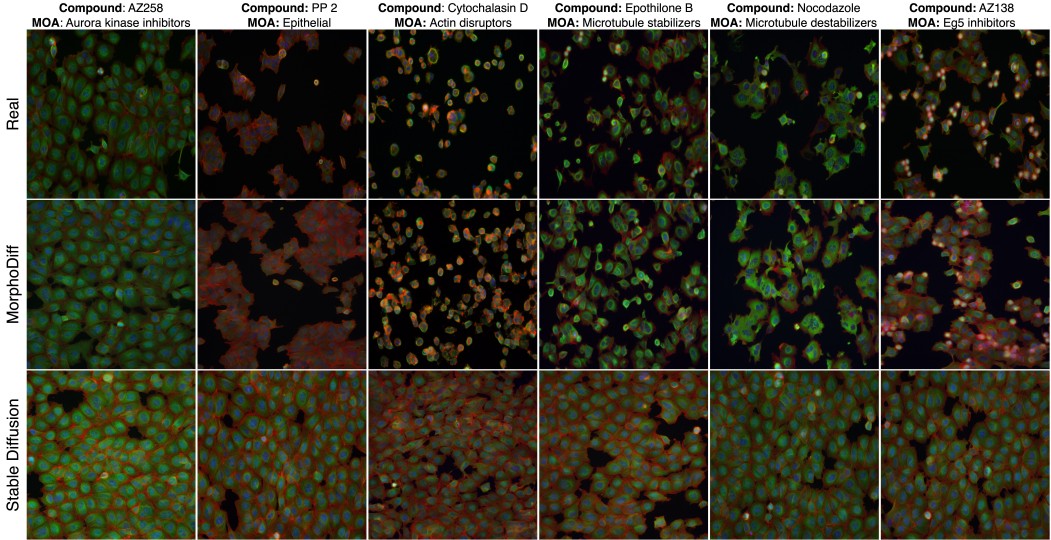

Figure 2: **MorphoDiff captures perturbation-specific cell morphology signals:** Sample images from the BBBC021 (14 Compounds) experiment demonstrating perturbation-induced cellular morphology for 6 compounds, each representing one of the annotated Mechanism of Actions (MOA) present in this experiment. The top row shows real images; the middle row displays MorphoDiff-generated images; and the bottom row presents images generated by unconditional SD model.

### 3.3.3 MORPHODIFF LEARNS BIOLOGICALLY INTERPRETABLE FEATURES

To validate the model's capacity to generate images with biologically meaningful representation, we expanded our benchmarking efforts using CellProfiler, a widely-used image analysis software in biomedical research to extract morphological cellular features from images across all conditions (Carpenter et al., 2006; Stirling et al., 2021). We quantified a comprehensive set of measurements related to cellular morphology across the different channels for the Cell, Nucleus, and Cytoplasm compartments. All features were pre-processed and standardized following best practices before analysis (Serrano et al., 2023). This approach allowed for a systematic comparison of CellProfiler-extracted features between different cell groups, providing a robust evaluation of the biological accuracy of generated images. Further description of CellProfiler feature extraction and processing are provided in the Appendix Note A.3.6.

In our initial validation, we applied Principal Component Analysis (PCA) to the pre-processed Cell-Profiler features derived from MorphoDiff-generated images, as shown in Figure 3a (3D visualizations are provided in the supplementary material). Sample distribution in the space indicated that biological features extracted from MorphoDiff-generated images under different compound treatments clustered cohesively in feature space. Additionally, when annotated by MOA labels visualized in Figure 3b, many compound groups with similar MOAs formed distinct clusters, reflecting their distinct biological effects. The MOA annotations are provided in Appendix Table 7.

We further applied K-Means clustering to CellProfiler features representing Cell, Nuclei, and Cytoplasm characteristics, using five different seeds to ensure robust evaluation. The clustering performance based on perturbation (compound) and MOA labels in real samples highlighted the inherent

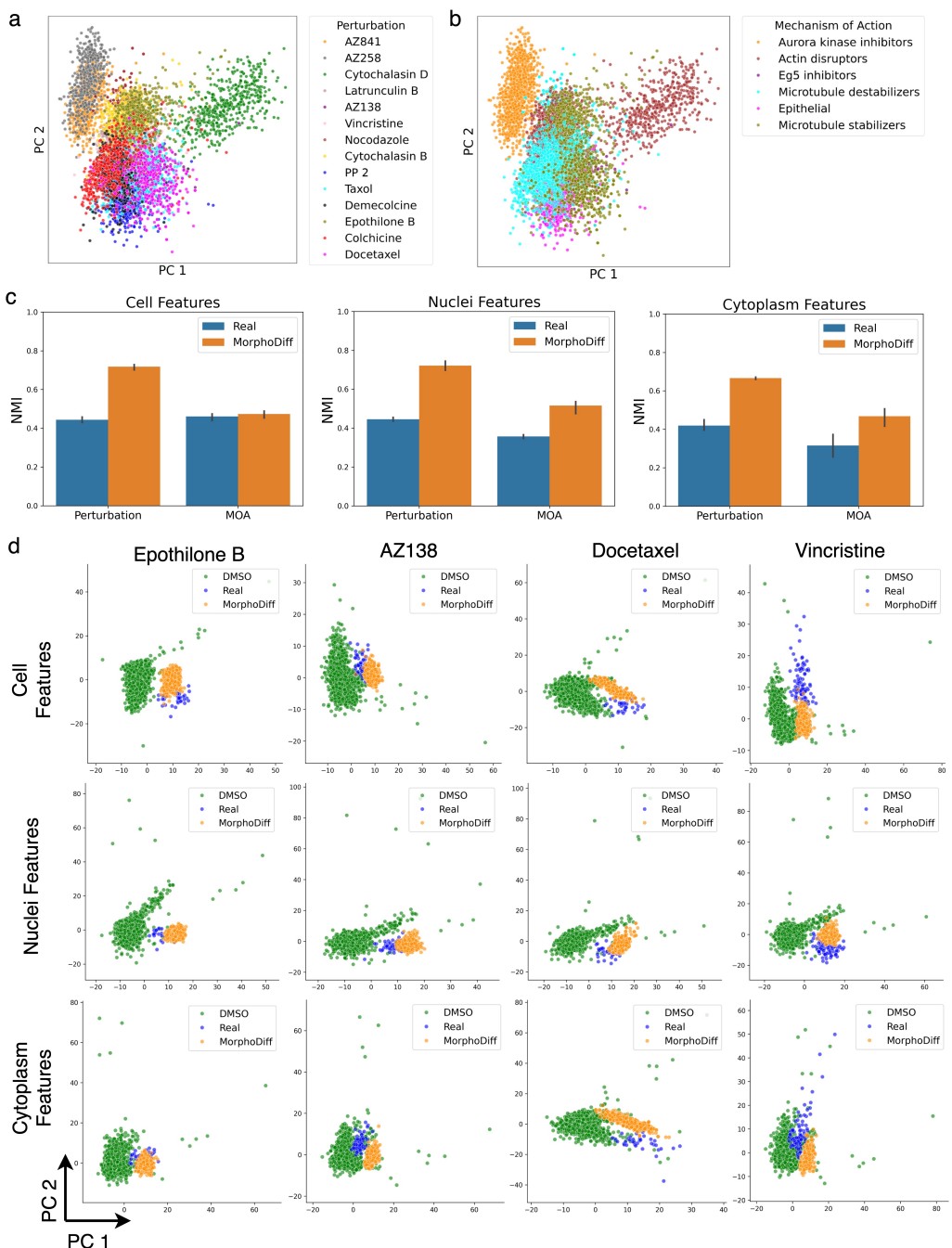

Figure 3: **Biologically interpretable feature analysis.** PCA visualization of CellProfiler features extracted from MorphoDiff-generated images, annotated by (a) compound labels and (b) Mechanism of Action (MOA) labels. (c) Normalized Mutual Information (NMI) metrics obtained from clustering CellProfiler features of real and MorphoDiff generated images (considering perturbation and MOA labels) using the K-Means algorithm with 5 different seeds, comparing their performance across three feature subsets representing Cell, Nuclei, and Cytoplasm properties. (d) Perturbation-specific PCA visualization of CellProfiler features comparing DMSO, real, and MorphoDiff-generated images for four example compounds for different feature subsets.

complexity of feature space, with overlapping sample distributions making clustering challenging. However, comparing the K-Means clustering results on the MorphoDiff-generated features summarized in 3c showed that the model maintained this complexity while improving the distinction between different compounds and MOAs, in most cases, which resulted in enhanced clustering performance. Overall, the visualizations and clustering analysis provide strong evidence that features extracted from MorphoDiff-generated images capture perturbation-specific and MOA-specific patterns, underscoring the model's ability to generate biologically differentiated sample groups. PCA and clustering analysis for the cropped images are provided in Appendix Figure 10.

We examined the distributional shift of CellProfiler features in MorphoDiff generated images compared to those extracted from real images from cells treated with the the same compound versus DMSO (control). Figure 3d illustrates sample distribution across three groups of CellProfiler features for four representative compounds. Notably, the generated samples' distribution consistently demonstrated a shift towards the feature distribution of real treated samples, diverging from DMSO controls. This trend provides strong evidence that MorphoDiff successfully learns biologically meaningful features that closely align with the target phenotypic space. Additional visualizations for these compounds, encompassing morphological features representing Area Shape, Texture, and a set of features that describe the shape of cells using a basis of Zernike polynomials (Zernike) are presented in Appendix Figure 11 (Yang et al., 2021). Furthermore, additional visualizations for all six feature subsets across a broader range of compounds and cropped version of images are provided in Appendix Figure 12 and 13, reinforcing the robustness of our findings.

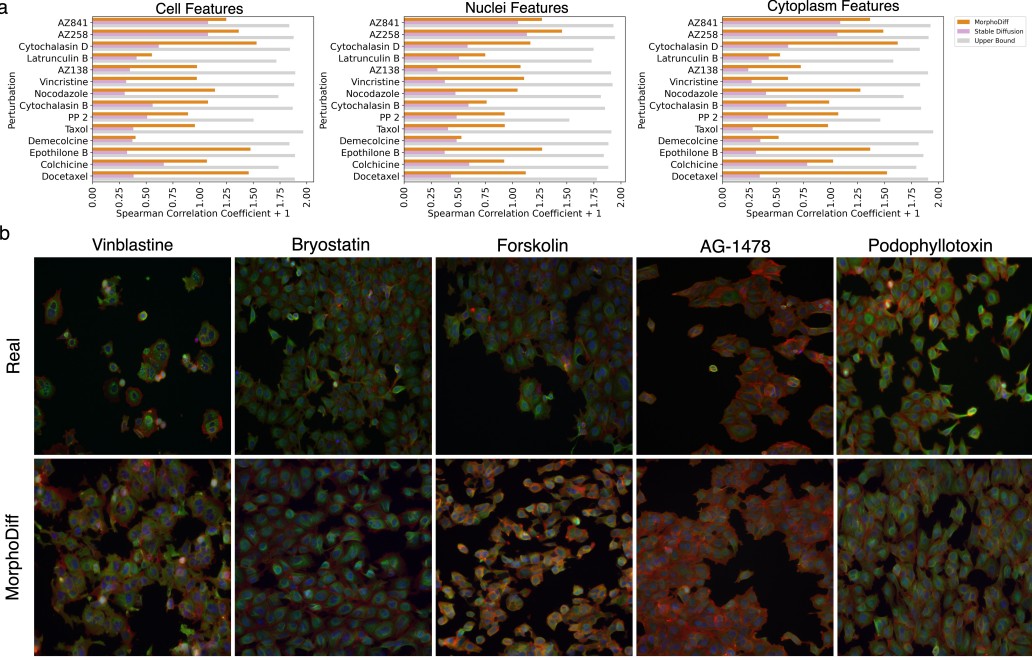

Figure 4: **MorphoDiff improves biological feature correlation and generalizes to unseen perturbations.** (a) Spearman correlation coefficient (+ 1) of CellProfiler features extracted from images generated by the generative models compared to the same features from real perturbed images across Cell, Nuclei, and Cytoplasm compartments. Upper bound correlation between random split of real image features are provided. (b) Sample images of the top 5 unseen compounds, comparing cellular phenotypes in real vs. MorphoDiff-generated cohorts.

To assess the models' ability to replicate cellular morphology, we calculated Spearman correlation coefficients between the averaged CellProfiler features of real and generated images for each compound. Given that feature correlations can vary across compounds due to factors like potency and morphological divergence from DMSO-treated controls, we focused on the improvement in positive

correlation achieved by MorphoDiff compared to the SD model. Figure 4a displays the correlation coefficients for Cell, Nuclei, and Cytoplasm features, which align with the range reported in previous work (Yang et al., 2021). The significant improvement observed in MorphoDiff over the baseline suggests it can generate biologically relevant, compound-specific cellular morphologies.

In addition to these major results, we noticed MorphoDiff images replicated certain biological nuances present in real images. We examined Pearson correlation coefficients between averaged CellProfiler features of images for pairs of 14 compounds in the BBBC021 experiment. Notably, the extreme cases of compound correlations observed in real images were faithfully reproduced in MorphoDiff-generated features. Specifically, AZ841 and AZ258 exhibited the highest correlation (0.96 in real vs 0.95 in MorphoDiff generated images), while AZ841 and Cytochalasin D demonstrated the lowest correlation ($-0.029$ in real vs. $-0.45$ in MorphoDiff generated images). The distributional similarity of AZ841 and AZ258 can also be observed in Figure 3a. Preservation of relative phenotypic relationships among compounds underscores MorphoDiff's capacity to capture nuanced biological interactions. Moreover, comparing standard-processed and cropped images in the BBBC021 experiments showed a slight improvement in computational metrics and sample distributions for cropped images overall (see Appendix Table 6), possibly due to enhanced compound-specific signals. While cropping may boost model performance, further research is needed to assess its impact on ML learning and its practicality.

### 3.3.4 MORPHODIFF GENERALIZABILITY TO UNSEEN DRUGS

We investigated MorphoDiff's capability to generalize to held-out compounds, an important yet challenging task in drug response prediction (Saha et al., 2024). Using Pearson correlation of compounds' structural embeddings generated by RDKit, we evaluated MorphoDiff's performance on the top 15 held-out compounds most correlated with in-distribution drugs (see Appendix Table 8), which allowed us to assess its generalization to unseen perturbations. Computational validation using FID and KID metrics revealed significant improvements in MorphoDiff-generated vs. SD-generated treated samples, with performance metrics summarized in Table 3 for both standard and cropped experiments. These results underscore MorphoDiff's capability in generalization as a diffusion-based model, which could be further enhanced with larger, more diverse training sets and advanced computational resources.

Table 3: FID ($\times 10^{-2}$) and KID metrics of real and generated images for unseen compounds. Bold values indicate p-value $< 0.05$, * indicates p-value $< 0.01$, and ** indicates p-value $< 0.001$

| Experiment | FID↓ (Standard) | FID↓ (Cropped) | KID↓ (Standard) | KID↓ (Cropped) |
|---|---|---|---|---|
| MorphoDiff | **1.83**** | **1.57*** | **0.2*** | **0.17** |
| Stable Diffusion | 2.56 | 2.03 | 0.30 | 0.23 |

Figure 4b shows images for the top five held-out drugs most correlated with in-distribution compounds. While most generated images closely resembled real samples based on molecular structure embeddings, Forskolin exhibited a distinct phenotype. Our analysis revealed these images were more similar to Cytochalasin D, Forskolin's most correlated in-distribution compound, suggesting that MorphoDiff generalizes to new compounds by leveraging perturbation embedding similarities. This highlights the impact of perturbation encoding accuracy on performance. While the current setting of MorphoDiff pipeline offers flexibility to integrate various tools, joint learning of the generative pipeline and perturbation encoders could enhance the model's generalizability.

## 4 CONCLUSION

We have introduced MorphoDiff, a new diffusion model-based generative framework for predicting cellular morphological responses to genetic and chemical perturbations in high-resolution images. Our experiments have demonstrated promising performance with respect to fidelity metrics and biological interpretability of the generated images. We expect that MorphoDiff can serve as a strong foundation for advancing cellular response prediction tools. Future work could incorporate additional covariates, such as drug concentration and cell type, by leveraging scaled imaging datasets and further investment in this area. Additionally, exploring combinatorial therapies and their synergistic effects presents an exciting direction toward advancing precision medicine.

## ACKNOWLEDGMENTS

This work was supported by Natural Sciences and Engineering Research Council of Canada Discovery Grant [RGPIN-2021-02680] (to B.H.K.), and funding from the NIH P41 GM135019 (to A.E.C. and B.A.C.).
We would like to thank Drs. Paula Llanos and Suganya Sivagurunathan from the Cimini Lab, as well as Dr. Ian Smith, for their valuable guidance and expertise in biological image analysis and data interpretation.

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

# A    APPENDIX

## A.1    FIGURES

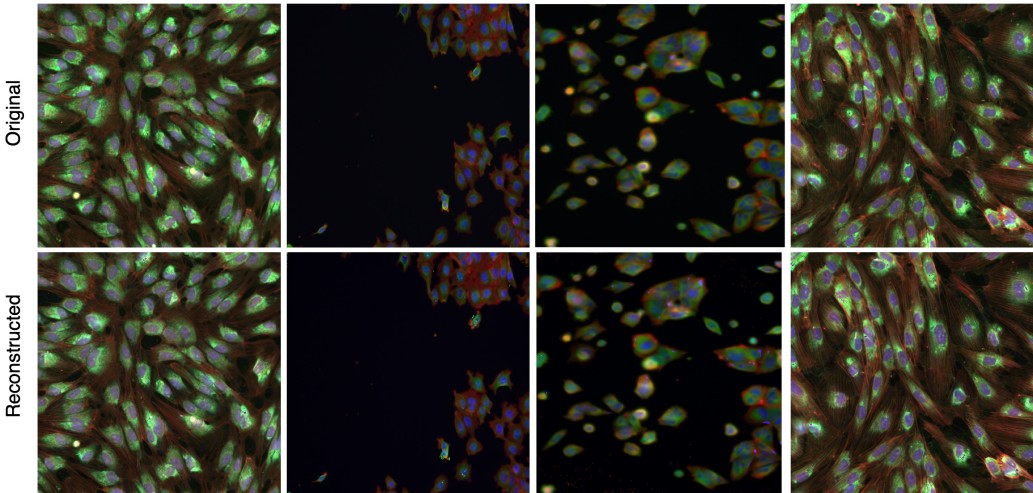

Figure 5: Sample images with their reconstructed version generated by the Stable Diffusion Variational Autoencoder module used for image encoding and decoding in the MorphoDiff pipeline.

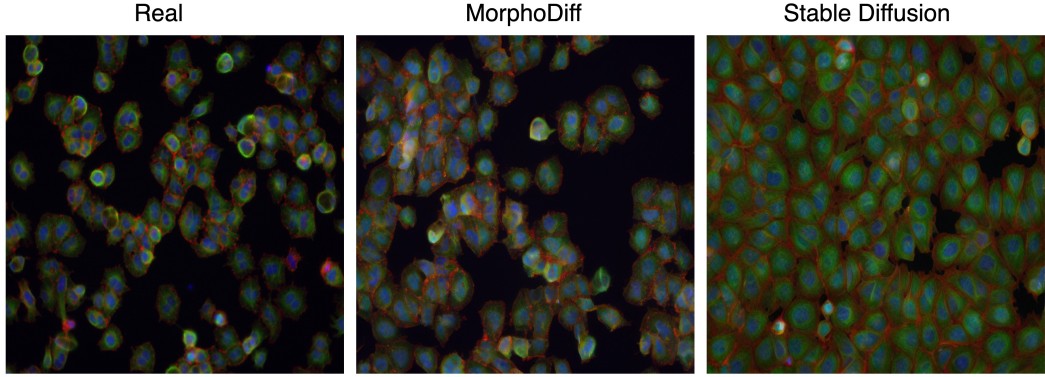

Figure 6: Random sample images generated by MorphoDiff, Stable Diffusion and from real cohorts for the Cytochalasin B compound.

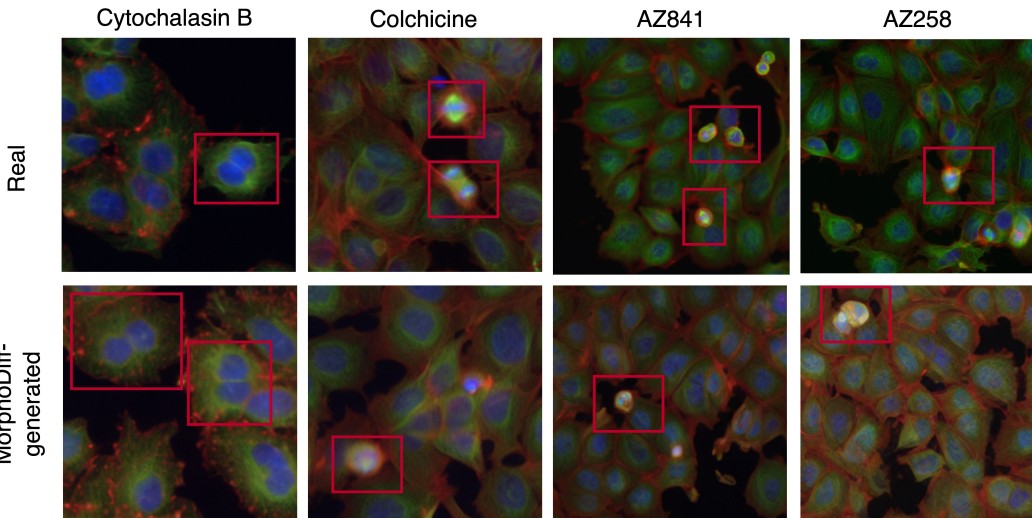

Figure 7: Random sample images generated by MorphoDiff and from real cohorts, demonstrating the generated images' capturing infrequent phenotypes associated with cell cycle alterations, such as bi-nucleated cells (in Cytochalasin B) and cells in metaphase stage (Colchicine, AZ841, and AZ258).

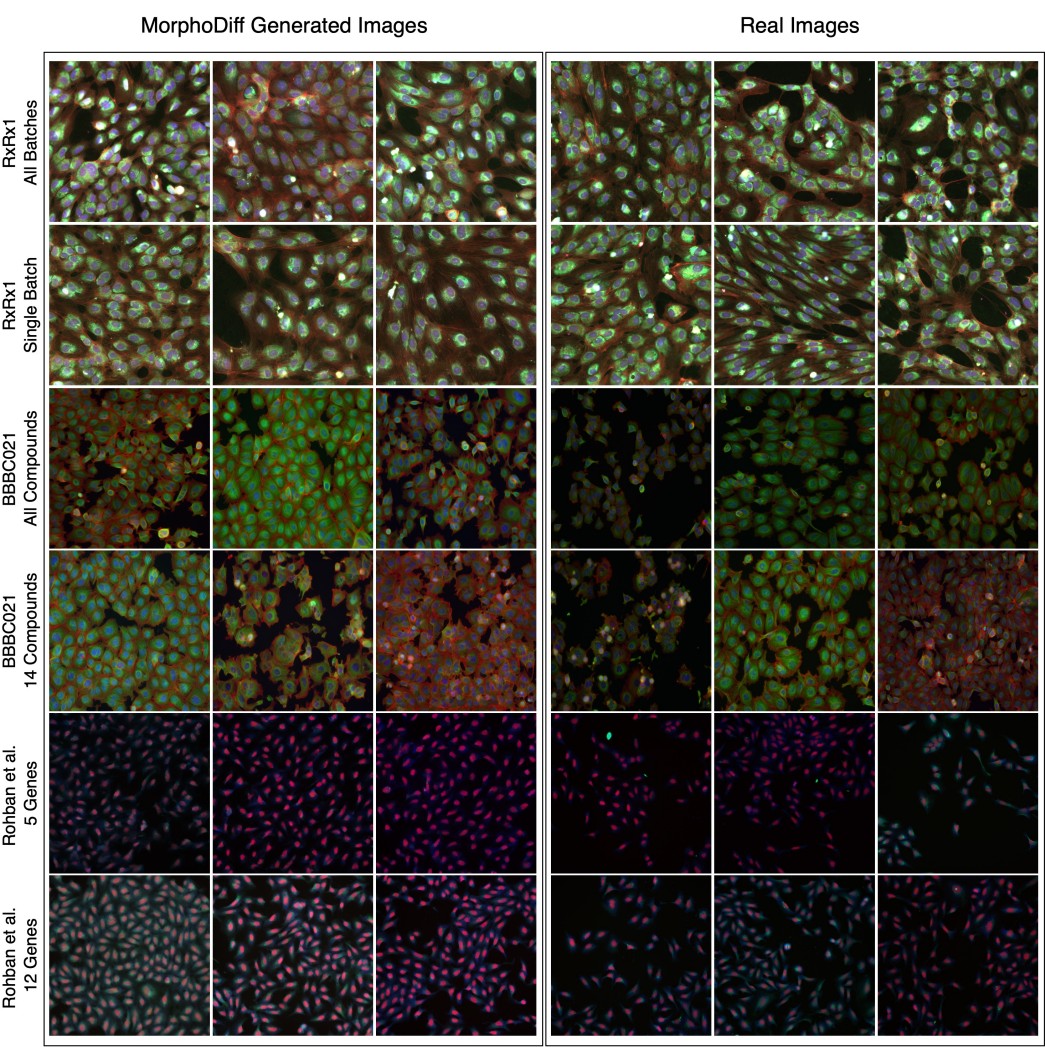

Figure 8: Random sample images generated by MorphoDiff and from real cohorts in different datasets.

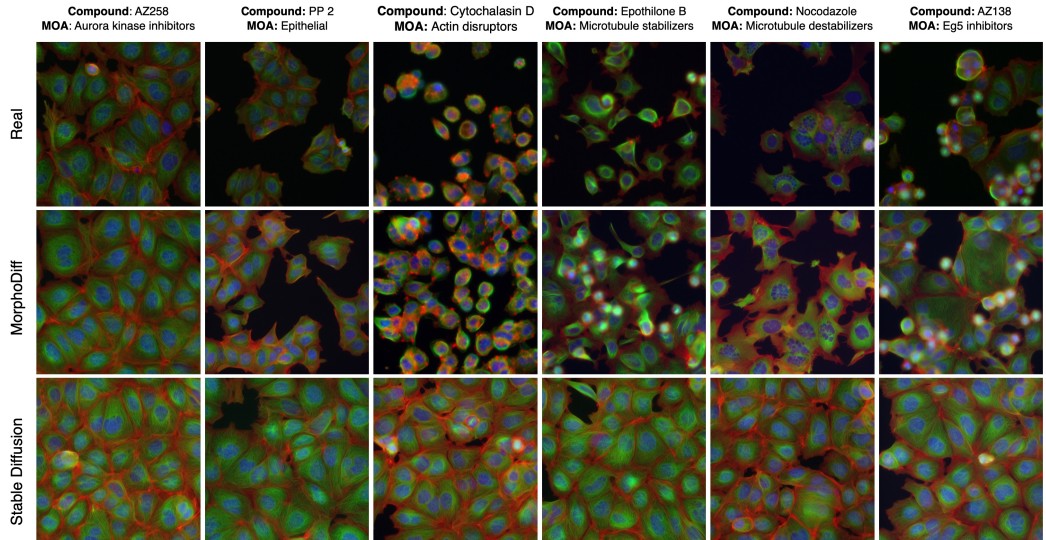

Figure 9: Sample images from the BBBC021 (14 Compounds) experiment with cropped pre-processing including smaller well area, demonstrating perturbation-induced cellular morphology for 6 compounds: AZ258, PP 2, Cytochalasin D, Epothilone B, Nocodazole, and AZ138, each representing one of the annotated Mechanism of Action (MOA) groups present in this dataset. The top row shows real (ground truth) images: the middle row displays images generated by the MorphoDiff pipeline: and the bottom row presents images generated by unconditional Stable Diffusion model.

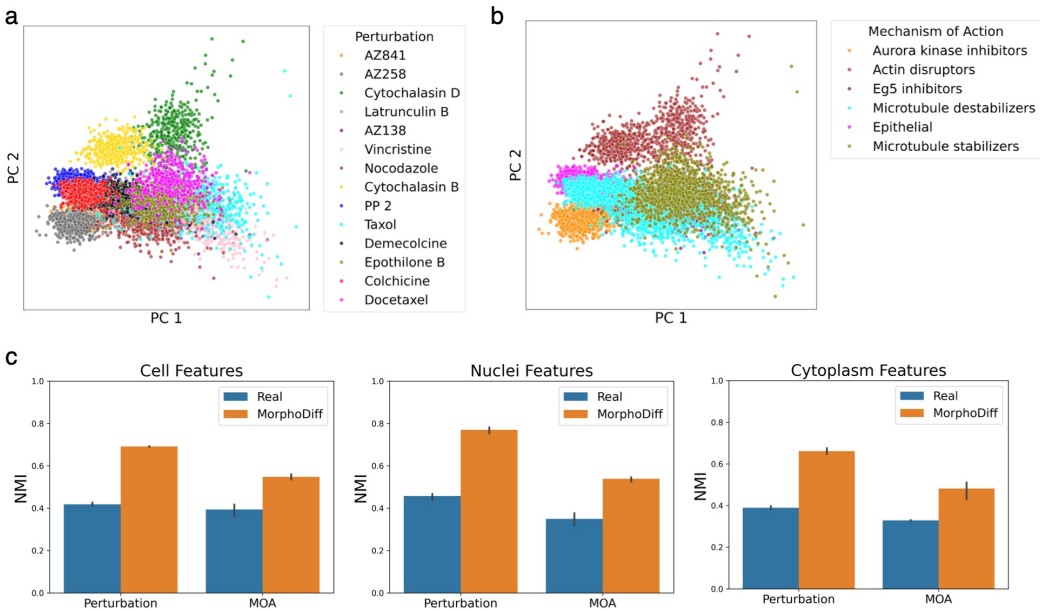

Figure 10: PCA visualization of CellProfiler features extracted from MorphoDiff-generated images in cropped image processing experiment of the BBBC021 experiment with 14 compounds, annotated by (a) compound labels and (b) Mechanism of Action (MOA) labels. (c) Normalized Mutual Information (NMI) metrics obtained from clustering CellProfiler features of real and MorphoDiff generated images using the K-Means algorithm with 5 different seeds, comparing their performance across three feature subsets representing Cell, Nuclei, and Cytoplasm properties.

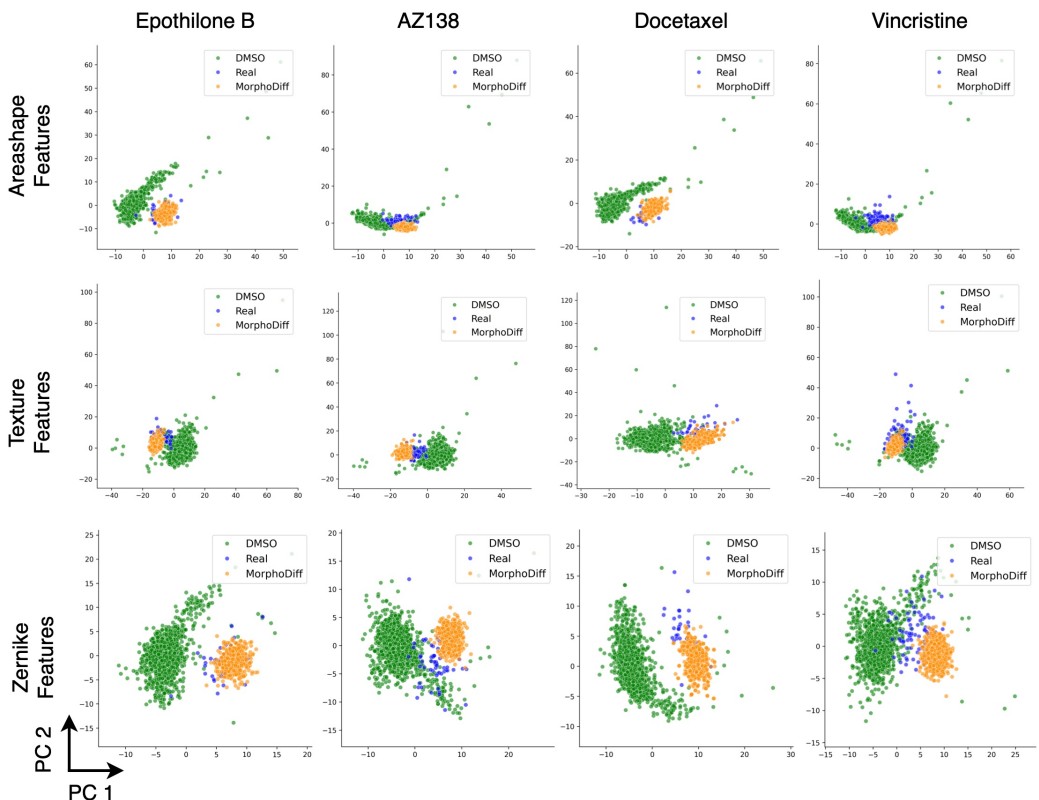

Figure 11: Perturbation-specific PCA visualization of CellProfiler features of the BBBC021 experiment with 14 compounds comparing DMSO, real perturbed, and MorphoDiff-generated images for four example compounds: Epothilone B, AZ138, Docetaxel, and Vincristine considering different subsets of CellProfiler features including Area Shape, Texture, and Zernike relevant properties.

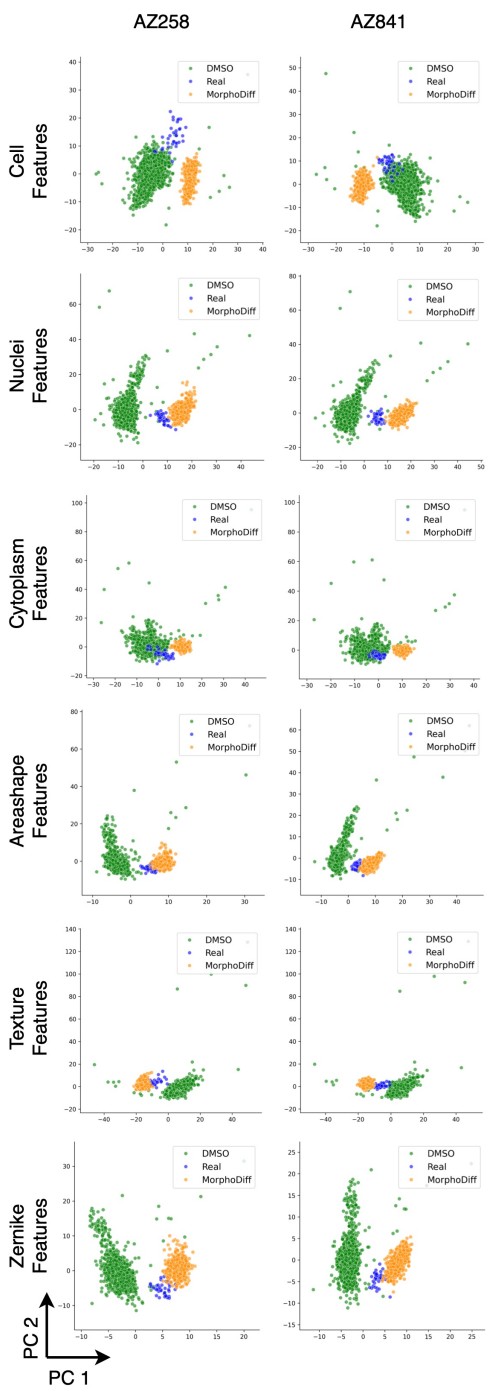

Figure 12: Perturbation-specific PCA visualization of CellProfiler features of the BBBC021 experiment with 14 compounds comparing DMSO, real perturbed, and MorphoDiff-generated images for two compounds: AZ258, AZ841, considering different subsets of CellProfiler features including Cell, Nuclei, Cytoplasm, Area Shape, Texture, and Zernike relevant properties.

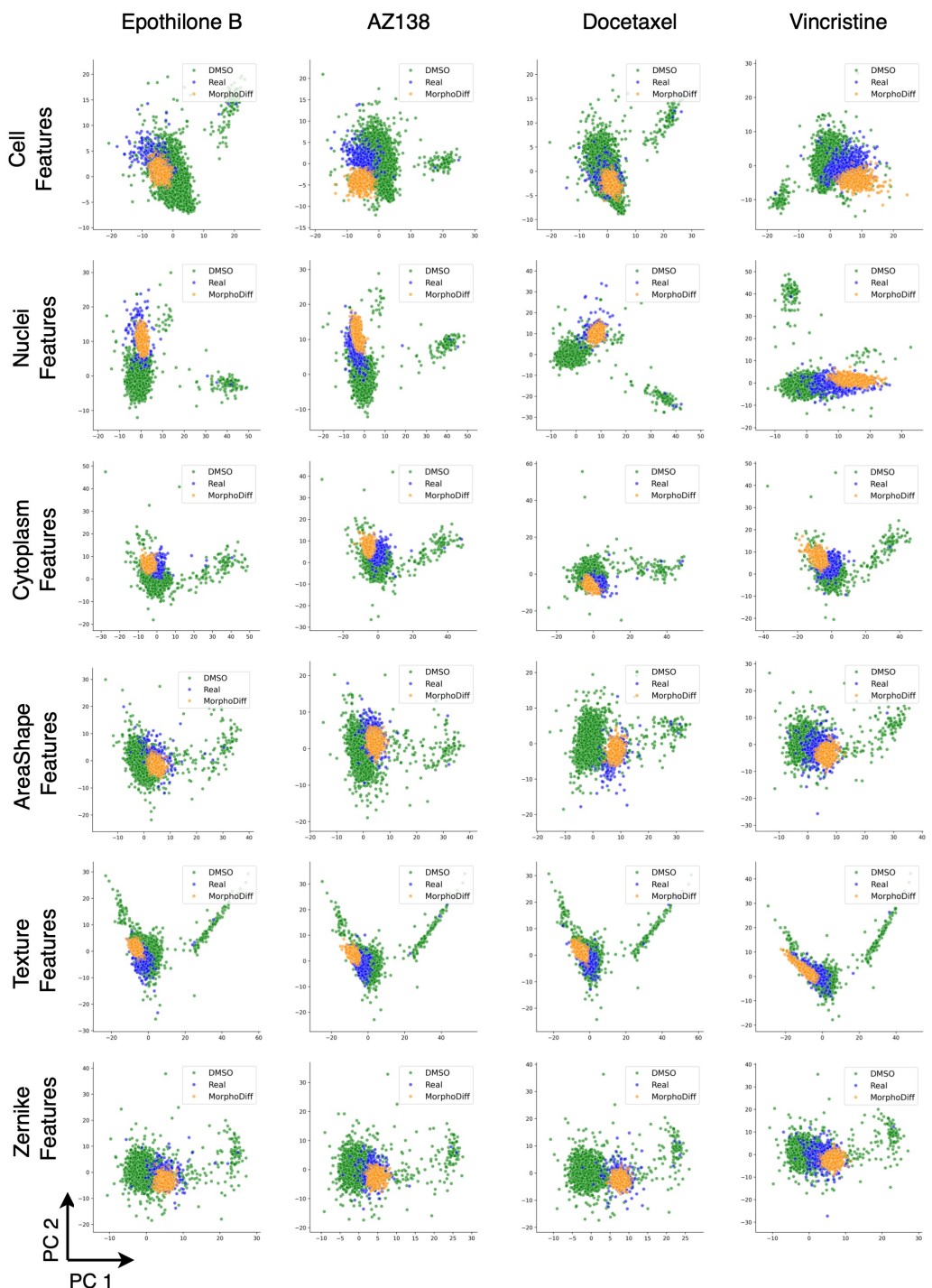

Figure 13: Perturbation-specific PCA visualization of CellProfiler features of cropped processed images in the BBBC021 experiment with 14 compounds, comparing DMSO, real perturbed, and MorphoDiff-generated images for four example compounds: Epothilone B, AZ138, Docetaxel, and Vincristine, considering different subsets of CellProfiler features including Cell, Nuclei, Cytoplasm, Area Shape, Texture, and Zernike relevant properties.

## A.2 TABLES

Table 4: Table of experiments' details from different datasets, including dataset name, experiment information, model type, number of trained steps, number of total training images, number of balanced images per perturbation included in training.

| Dataset | Experiment | Model | Trained steps | Training images | Per perturbation balanced images |
|---|---|---|---|---|---|
| RxRx1 | All Batches | MorphoDiff | [130K-133K] | 59K | $\sim 50$ |
| RxRx1 | All Batches | Stable Diffusion | [130K-133K] | 59K | $\sim 50$ |
| RxRx1 | Single Batch | MorphoDiff | [113K-118K] | 2.46K | $\sim 50$ |
| RxRx1 | Single Batch | Stable Diffusion | [113K-118K] | 2.46K | $\sim 50$ |
| BBBC021 | All Compounds | MorphoDiff | [140K-142K] | 98K | 1K |
| BBBC021 | All Compounds | Stable Diffusion | [140K-142K] | 98K | 1K |
| BBBC021 | 14 Compounds | MorphoDiff | [116K-120K] | 14K | 1K |
| BBBC021 | 14 Compounds | Stable Diffusion | [116K-120K] | 14K | 1K |
| Rohban et al. | 5 Genes | MorphoDiff | [130K] | 1K | 200 |
| Rohban et al. | 5 Genes | Stable Diffusion | [130K] | 1K | 200 |
| Rohban et al. | 12 Genes | MorphoDiff | [61K-63K] | 6K | 500 |
| Rohban et al. | 12 Genes | Stable Diffusion | [61K-63K] | 6K | 500 |

Table 5: Table showing the average rank of matched compounds based on FID and KID metric comparisons between generated images and all real perturbed image cohorts. Ranks were normalized by the total number of compounds in each experiment, with MorphoDiff consistently achieving a lower average rank compared to the unconditional Stable Diffusion model. A lower rank (highlighted in bold) indicates better performance, with the ideal case (rank 1st) representing the scenario where the generated images for a specific perturbation have the smallest distance metric with the matched real cohort.

| Dataset | Experiment | Method | FID↓ | KID↓ |
|---|---|---|---|---|
| RxRx1 | All Batches | MorphoDiff | **0.13** | **0.12** |
| RxRx1 | All Batches | Stable Diffusion | 0.51 | 0.51 |
| RxRx1 | Single Batch | MorphoDiff | **0.12** | **0.12** |
| RxRx1 | Single Batch | Stable Diffusion | 0.51 | 0.51 |
| BBBC021 | All Compounds | MorphoDiff | **0.41** | **0.42** |
| BBBC021 | All Compounds | Stable Diffusion | 0.50 | 0.50 |
| BBBC021 | 14 Compounds | MorphoDiff | **0.42** | **0.43** |
| BBBC021 | 14 Compounds | Stable Diffusion | 0.53 | 0.53 |
| Rohban et al. | 5 Genes | MorphoDiff | **0.56** | **0.56** |
| Rohban et al. | 5 Genes | Stable Diffusion | 0.6 | 0.6 |
| Rohban et al. | 12 Genes | MorphoDiff | **0.51** | **0.5** |
| Rohban et al. | 12 Genes | Stable Diffusion | 0.54 | 0.54 |

Table 6: Average of FID ($\times 10^{-2}$) and KID metrics across all perturbations in the BBBC021 experiments for two different pre-processing approaches (standard and cropped around smaller well area), assessing distributional similarity between generated and real images from the corresponding perturbation conditions (lower is better). Bold values highlight statistically significant differences with p-value $< 0.05$, * indicates p-value $< 0.01$, and ** indicates p-value $< 0.001$.

| Dataset | Experiment | Method | FID↓ | KID↓ |
|---|---|---|---|---|
| BBBC021 | All Compounds (Standard) | MorphoDiff | **1.99**** | **0.21**** |
| BBBC021 | All Compounds (Standard) | Stable Diffusion | 3.84 | 0.47 |
| BBBC021 | All Compounds (Cropped) | MorphoDiff | **1.55**** | **0.14**** |
| BBBC021 | All Compounds (Cropped) | Stable Diffusion | 2.80 | 0.29 |
| BBBC021 | 14 Compounds (Standard) | MorphoDiff | **2.26*** | **0.30** |
| BBBC021 | 14 Compounds (Standard) | Stable Diffusion | 3.22 | 0.42 |
| BBBC021 | 14 Compounds (Cropped) | MorphoDiff | **1.48**** | **0.17**** |
| BBBC021 | 14 Compounds (Cropped) | Stable Diffusion | 2.42 | 0.30 |

Table 7: List of compounds and their annotated mechanism of actions (MOA) in the BBBC021 experiment (14 compounds)

| Compound | Mechanism Of Action (MOA) |
|---|---|
| AZ138 | Eg5 inhibitors |
| AZ841 | Aurora kinase inhibitors |
| AZ258 | Aurora kinase inhibitors |
| Cytochalasin B | Actin disruptors |
| Cytochalasin D | Actin disruptors |
| Latrunculin B | Actin disruptors |
| PP 2 | Epithelial |
| Demecolcine | Microtubule destabilizers |
| Nocodazole | Microtubule destabilizers |
| Colchicine | Microtubule destabilizers |
| Vincristine | Microtubule destabilizers |
| Epothilone B | Microtubule stabilizers |
| Taxol | Microtubule stabilizers |
| Docetaxel | Microtubule stabilizers |

Table 8: List of top most correlated held-out compounds to the in-distribution compounds (Based on the molecular encoding obtained from RDKit tool) in the BBBC021 experiment with 14 compounds included in training.

| In-distribution compound | Held-out compound | Pearson correlation |
|---|---|---|
| Vincristine | Vinblastine | 0.92 |
| PP 2 | AG-1478 | 0.78 |
| Docetaxel | Bryostatin | 0.70 |
| Colchicine | Podophyllotoxin | 0.70 |
| Taxol | Bryostatin | 0.66 |
| Cytochalasin D | Forskolin | 0.61 |
| Cytochalasin D | Rapamycin | 0.61 |
| Demecolcine | Emetine | 0.57 |
| PP 2 | AZ701 | 0.53 |
| Cytochalasin B | Simvastatin | 0.52 |
| Cytochalasin B | Mevinolin-lovastatin | 0.50 |
| Docetaxel | Nystatin | 0.49 |
| Nocodazole | Acyclovir | 0.48 |
| PP 2 | Bohemine | 0.47 |
| PP 2 | Roscovitine | 0.46 |
| Cytochalasin B | Taurocholate | 0.46 |

### A.3 NOTES

#### A.3.1 PERTURBATION ENCODING

MorphoDiff offers the flexibility to incorporate different perturbation encoding modules, tailored to the type and properties of the perturbation.

For genetic perturbations, gene IDs from the original dataset were mapped to genes available in the scGPT model, pretrained on 33 million normal human cells[2]. The 512-dimensional gene embeddings generated by this model were incorporated into the MorphoDiff pipeline as model conditions. For the RxRx1 dataset, siRNA perturbation IDs were mapped to their associated gene IDs using the Thermo Fisher Scientific website[3]. In the dataset by Rohban et al., gene IDs were directly provided as the perturbations associated with each treated image.

For compound perturbations in the BBBC021 dataset, SMILES representations of each compound were provided as metadata by the Broad Institute website[4]. RDKit software was applied to the SMILES representations to generate numerical encodings representing chemical structures, with each feature normalized by its mean and standard deviation.

The code for generating chemical and genetic perturbation embeddings is available in the project's GitHub.

#### A.3.2 TECHNICAL DETAILS

Publicly released parameters of Stable Diffusion v1.4[5] along with the accompanying training script from Huggingface were used as the base model[6]. Perturbation embeddings generated by the projection module were padded to match the shape of Stable Diffusion's prompt embedding. During training, we applied the Exponential Moving Average (EMA) technique to enhance model learning and stability (Tarvainen & Valpola, 2017). EMA is employed to reduce noise during training and improve the model's generalization performance. For all experiments, we used the most recent parameters available at the time of evaluation, and training was conducted based on the available computing and time resources, ensuring that the number of training steps across models for each experiment was comparable to facilitate a fair evaluation of models. To address class imbalance, we applied image augmentation techniques such as rotation and flipping on the training samples from each perturbation cohort, enhancing sample diversity. The code for data pre-processing, model training, evaluation, and result reproduction, along with the model weights are available on the project's GitHub page. All training and evaluation processes were conducted on NVIDIA T4 and NVIDIA A40 GPUs. Most hyper-parameters remained consistent across all models and were aligned with the original Stable Diffusion settings, unless stated. A batch size of 32 and a learning rate of $1 \times 10^{-5}$ were used for all training runs.

#### A.3.3 DATASETS

**RxRx1 dataset** developed by Recursion (Sypetkowski et al., 2023), contains 1108 perturbation classes across four distinct cell types. Each sample comprises six-channel fluorescent microscopy images capturing key cellular structures, including the nucleus, endoplasmic reticulum, actin cytoskeleton, nucleolus, mitochondria, and Golgi apparatus. Our analysis concentrated on the Human Umbilical Vein Endothelial Cells (HUVEC), the largest group within the dataset, comprising approximately 59,000 samples drawn from multiple experimental batches. We included only images with gene annotations available on the Thermo Fisher Scientific website[7], corresponding to the original siRNA perturbations. The RxRx1 authors identified pronounced batch effects across experimental runs, presenting a challenge for ML model training. To investigate this, we conducted two sets of experiments: one using all HUVEC images (RxRx1 - All Batches), and the other constrained to a single batch (RxRx1 - Single Batch) to minimize batch effects. Given the large number of

---

[2]https://github.com/bowang-lab/scGPT/tree/main

[3]https://www.thermofisher.com/ca/en/home.html

[4]Link to BBBC021 metadata

[5]https://huggingface.co/CompVis/stable-diffusion-v1-4

[6]Training Script

[7]https://www.thermofisher.com/ca/en/home.html

siRNA perturbations in RxRx1, a random subset of 50 siRNAs was selected for validation in each experiment, balancing the constraints of time and computational resources. The authors' provided code was used to convert the six-channel images into RGB format for input into the MorphoDiff and Stable Diffusion pipelines[8]. The images were directly downloaded from Recursion's website[9].

We further extended our analysis to the **BBBC021 dataset** comprising 13200 images of MCF7 breast cancer cells, stained for DNA, F-actin, and B-tubulin, and imaged using three-channel fluorescent microscopy (Caie et al., 2010). For consistency, these channels were directly mapped to RGB format for modelling and validation. The dataset profiles MCF7 cells treated with 113 small molecules, each administered at eight different concentrations over a 24-hour period. We conducted two sets of experiments: one using all compounds for which SMILES-based projections were available and the embedding was generated by the projection module (98 in total), and another included 14 compounds with 6 MOAs provided on the dataset website (Caie et al., 2010). These MOAs included Eg5 inhibitors, Aurora kinase inhibitors, Actin disruptors, Epithelial modulators, and Microtubule destabilizers, and Microtubule stabilizers (Caie et al., 2010). Images from this dataset were directly downloaded from Broad Bioimage Benchmark Collection website[10].

The **Rohban et al. dataset** contains Cell Painting images of U2OS cells with 323 over-expressed genes (Rohban et al., 2017). Based on expert consultation, we selected three of the five imaging channels—RNA, Mitochondria, and DNA—for modeling and validation, prioritizing channels essential for image segmentation and feature extraction that are also biologically informative and interpretable (Stirling et al., 2021; Carpenter et al., 2006). The choice of channels can be flexible depending on dataset properties. Two gene subsets were used from this dataset for modeling. In the first experiment, we used a set of five genes (gene list: RAC1, KRAS, CDC42, RHOA, PAK1) reported in the original study as being involved in pathways known to affect cellular morphology (Rohban et al., 2017). For the second experiment, we focused on gene clusters identified in (Rohban et al., 2017) based on morphological profiling. We selected genes from clusters with at least three genes, having a Gene Ontology (GO) term with a p-value below 0.01, and where at least half of the genes in the cluster were associated with the GO terms (gene list: XBP1, MAPK14, RAC1, AKT1, AKT3, RHOA, PRKACA, SMAD4, RPS6KB1, KRAS, BRAF, RAF1). These genes were sourced from Supplementary File 1F in (Rohban et al., 2017). Images for this dataset were directly downloaded form the Cell Painting Gallery GitHub page[11].

### A.3.4    PRE-PROCESSING

**Standard Pre-processing:** For all experiments, images larger than $512 \times 512$ pixels were resized to this dimension, preserving as much of the original image area as possible. Pre-processing followed Stable Diffusion best practices, scaling pixel values to the $(0, 1)$ range and normalizing with a mean and standard deviation of $0.5$ (Rombach et al., 2022).

**Cropped Pre-processing:** To evaluate the impact of focusing on smaller well areas with enhanced cellular texture, five $512 \times 512$ pixel patches were cropped from the four corners and center of the BBBC021 images. This approach augmented the data, and the processed images were different from the standard processing by magnifying cellular details. Cropped images were used for modeling and validation, with comparisons made against the standard pre-processing results. The same normalization process as in standard pre-processing was applied prior to further analysis.

**Cell Painting Specific Pre-processing:** For the RxRx1 and BBBC021 datasets, self-standardization was applied as recommended by (Sypetkowski et al., 2023), where each image channel was standardized separately. In the experiments from Rohban et al., illumination correction arrays provided by the authors were used to adjust for brightness differences as part of the pre-processing pipeline (Rohban et al., 2017).

---

[8]Conversion Code

[9]https://www.rxrx.ai/rxrx1

[10]https://bbbc.broadinstitute.org/BBBC021

[11]https://github.com/broadinstitute/cellpainting-gallery

### A.3.5 VALIDATION

MorphoDiff's performance was validated with respect to computational distance metrics, visual assessment, and CellProfiler feature analysis to interpret the biological relevance of the generated images. We also benchmarked its performance against the unconditional Stable Diffusion framework as the baseline by fixing the input prompt condition, and fine-tuning the SD framework on Cell Painting images, similar to what was used to train MorphoDiff.

For benchamrking on cell cropped patches, we used the IMPA checkpoint trained on the BBBC021 dataset (six conditions: DMSO and five compounds)[12]. Our models, trained on 14 compounds, were adapted for image quality evaluation. Using a pre-trained generalist cell segmentation model[13], we segmented generated and real images to obtain cell masks. Consistent $96 \times 96$ pixel patches centered around single cells were cropped for fair evaluation against IMPA images, and compound-specific FID and KID scores were calculated. Following the Comparison section in the main manuscript and to provide a robust evaluation, 500 cell patches were randomly selected for ground truth and the same amount were generated for each model (Stable Diffusion, IMPA, and MorphoDiff) across ten random seeds, with image quality metrics and significance tests reported for each compound.

In the CellProfiler analysis, image-level features extracted from real images were compared to those extracted from the generated images. Notably, the number of real images per condition was often smaller than the number of generated images. The two-sample t-test was used for statistical analysis, with p-values of less than $0.05$ were considered statistically significant unless otherwise stated.

### A.3.6 CELLPROFILER ANALYSIS

We used the CellProfiler tool for quantitative phenotype measurement across thousands of images, as it is widely regarded as the state-of-the-art image analysis software in the literature. Due to the large volume of images, we employed Distributed CellProfiler (McQuin et al., 2018), which facilitates running a Dockerized version of CellProfiler on Amazon Web Services (AWS) by leveraging AWS's storage and computing infrastructure. All CellProfiler pipelines used for analysis were implemented and customized by experts for each experiment.

A comprehensive set of cellular morphology measurements was extracted across different channels for the Cell, Nucleus, and Cytoplasm compartments. The mean, median, and standard deviation of extracted features were summarized for each image. Image features were pre-processed following best practices, including the removal of features containing NaNs, elimination of outliers, and removal of highly correlated features. All features were normalized prior to analysis using the "standardized" method from pycytominer Python package[14] (Serrano et al., 2023). We analyzed different subsets of CellProfiler features (1088 in total) by focusing on those included "Cell" (300 features), "Nuclei" (394 features), "Cytoplasm" (318 features), "AreaShape" (267 features), "Texture" (346 feature) and "Zernike" (225 features) in their feature names, each correspond to aspects of cell morphological properties. The CellProfiler pipeline, feature processing and analysis scripts are provided on the project's GitHub page.

---

[12]https://github.com/theislab/IMPA
[13]https://github.com/Lee-Gihun/MEDIAR
[14]https://github.com/cytomining/pycytominer

