# OpenReview forum: "MorphoDiff: Cellular Morphology Painting with Diffusion Models"
_ICLR.cc/2025/Conference — ICLR 2025 Spotlight_

### Official Review · Reviewer_qrds · 2024-10-30

**Soundness:** 3
**Presentation:** 4
**Contribution:** 3
**Rating:** 8
**Confidence:** 4

**Summary:**

The authors propose MorphoDiff, a perturbation-based conditional latent diffusion model designed to generate high-resolution images of drug-treated cell morphologies. The model incorporates both chemical and genetic perturbations as conditioning inputs, enabling it to generalise across diverse interventions. Evaluations indicate that MorphoDiff exceeds the performance of a conventional Stable Diffusion model in replicating biologically realistic cellular responses, with strong results in both visual fidelity and biological interpretability.

**Strengths:**

## Clear Structure and Motivating Background:
The paper is well-organised, with each section contributing logically to the overall narrative. The background provides a thorough review of core concepts in conditional diffusion modeling, highlighting the advantages of diffusion models for image generation tasks in biological contexts.

## Innovative Use of scGPT for Perturbation Conditioning:
This integration of a large, pre-trained transformer model to embed meaningful genetic perturbation features enables MorphoDiff to generalise across complex perturbations with improved accuracy, potentially opening new avenues for generative applications in cellular biology.

## Comprehensive Biological Validation Beyond Standard Generative Metrics:
The authors go beyond typical generative metrics such as FID by evaluating MorphoDiff’s performance on biological realism through CellProfiler, a tool widely used in biological research. This thorough approach highlights the biological validity of generated samples by extracting quantitative morphological features, offering a layer of interpretability and trustworthiness that is essential for practical applications in drug discovery. The use of CellProfiler also strengthens the model’s credibility by demonstrating the relevance of the generated images in a real biological context, not just in terms of visual fidelity.

**Weaknesses:**

## Limited Range of Comparative Baselines:
While MorphoDiff is evaluated in terms of biological interpretability and generative quality, the paper lacks a sufficient range of comparative baselines, particularly given the advancements in conditional latent diffusion models. The authors mention MO2Image, yet they do not compare their approach to several other conditional diffusion models that could have served as relevant baselines. For instance, competing methods like PhenDiff [1] offer similar approaches in generating drug-treated cellular images using conditional diffusion. Expanding the range of baselines would solidify MorphoDiff’s competitive advantage and contextualise its performance relative to other SOTA models.

## Clarification on Novelty Claims:
The claim that MorphoDiff is the first to handle both chemical and genetic perturbations in a conditional latent diffusion model is somewhat overstated, as methods like PhenDiff [1] have previously addressed similar challenges. Citing these comparable methods (or similar) and clearly articulating MorphoDiff’s unique contributions (e.g., specific model adaptations, or validation approach) would improve the transparency of the novelty claim.

## Insufficient Detail in Key Methodological Components:
Certain methodological sectionsc lack sufficient depth, making it difficult to reproduce the model. The perturbation projection module, which appears to be a crucial element of MorphoDiff’s design, is only briefly discussed. Given that this module enables MorphoDiff to conditionally generate diverse morphological responses, more information on how it was adapted for this task would enhance clarity and allow for reproducibility. Furthermore, the absence of code or a repository also hinders reproducibility.

## Overlooked Metrics for Sample Variability:
The paper does not address generated sample variability or uniqueness within the output set for each condition. Evaluating generated sample uniqueness is particularly important in high-fidelity generation, as latent diffusion models can sometimes suffer from mode collapse, resulting in overly similar or even identical samples across iterations. Implementing additional metrics such as Structural Similarity Index (SSIM) could quantify the variability of generated samples within each condition, helping to ensure MorphoDiff generates a diverse range of morphologies. These metrics would also provide insight into the model's robustness and its ability to capture subtle differences in morphological responses across varying treatments.


[1] https://arxiv.org/pdf/2312.08290

**Questions:**

## Sensitivity to Diffusion Model Selection:
How sensitive is MorphoDiff to the choice of latent diffusion model? The authors use Stable Diffusion as a baseline, but further exploration into alternative pre-trained diffusion models could provide valuable insights.

---

> ### Author Response · Authors · 2024-11-23
>
> We thank the reviewer for their kind remarks on the contributions of our work. Please find below our responses to the raised questions.
>
> ---
> **Regarding “Limited Range of Comparative Baselines”:**
>
> We appreciate the reviewer’s suggestions to include benchmarking against other potential baselines and improve validation experiments. We fully recognize the importance of comparing our model to other published methods to provide a comprehensive evaluation of its capabilities, as well as the need for meaningful validation to motivate realistic applications. Below, we outline our efforts to address these concerns:
>
> 1. **Comparison with Mol2Image:**
>
>   We identified the Mol2Image paper as the most relevant to our pipeline in terms of input/output parameters and image scale. We tried our best to add this comparison to the experiments. The authors provided model weights for a dataset they used in their study [1]. However, no script or guideline are provided for training or fine-tuning to validate their method on other datasets. We also reached out multiple times to the authors directly but have not received any response and were unable to obtain the necessary material for adapting their model to our experiments and perform benchmarking.
>
> 2. **Comparison with IMPA and PhenDiff:**
>
>    We have successfully added benchmarking of MorphoDiff against the IMPA model [2] using their provided checkpoint for the BBBC021 dataset (trained on six conditions: five compounds and DMSO, provided in https://zenodo.org/record/8307629). Our models, trained on 14 compounds, were adapted for image quality evaluation. Using a pre-trained generalist cell segmentation model (https://github.com/Lee-Gihun/MEDIAR), we segmented generated and real images to obtain cell masks. Consistent 96×96 pixel patches centered around single cells (with IMPA images) were cropped for fair evaluation against IMPA images, and compound-specific FID and KID scores were calculated. Following the Comparison section in the main manuscript and to provide a robust evaluation, 500 cell patches were randomly selected for ground truth and the same amount were generated for each model (Stable Diffusion, IMPA, and MorphoDiff) across ten random seeds, with image quality metrics and significance tests reported for each compound as provided in the below table.
>
>
> | Model            | AZ138 (FID/KID) | AZ258 (FID/KID) | Taxol (FID/KID) | Cytochalasin B (FID/KID) | Vincristine (FID/KID) |
> |-------------------|-----------------|-----------------|-----------------|--------------------------|-----------------------|
> | Stable Diffusion | 1.40 / 0.14     | 0.94 / 0.08     | 1.50 / 0.15     | **0.83 / 0.07**          | 1.83 / 0.20          |
> | IMPA             | 0.98 / 0.07     | 1.18 / 0.12     | 1.28 / 0.11     | 1.23 / 0.11              | 1.05 / 0.07          |
> | MorphoDiff       | **0.82 / 0.06** | **0.76 / 0.05** | **1.09 / 0.10** | 1.19 / 0.11              | **0.86 / 0.06**      |
>
>
> Statistical testing (t-tests between the best and the second best method) demonstrated that MorphoDiff outperforms all other methods across all compounds, except for Cytochalasin B. Our investigation showed that high resolution images of Cytochalasin B treated images look more similar to real perturbed images for this drug compared to Stable Diffusion. The obtained FID and KID metrics in high resolution scale also showed MorphoDiff outperforms Stable Diffusion (both with standard preprocessing and cropped preprocessing), with sample images provided in the newly added Appendix Figure 6). We believe generating and validation of images with larger fields of view enables capturing cellular density and intercellular relationships, which will provide a more comprehensive insight of the cellular interactions and phenotypic shifts induced by different perturbations, as well as cellular diversity within the well. Detailed explanations of this analysis are provided in Appendix Note Validation subsection.
>
> For the PhenDiff model [3], we unfortunately encountered challenges in reproducing it due to the absence of available checkpoints and detailed guidelines for conditional data preparation. Despite reaching out to the authors, we have not received further information to facilitate reproducing their results.
>
> We hope this additional validation further demonstrates MorphoDiff's potential to advance virtual phenotypic screening tasks. Thank you for emphasizing this critical aspect, which has helped substantially enhance the strength of our paper.

---

> > ### Author Response · Authors · 2024-11-23
> >
> > References for the previous comment:
> >
> > [1] Hofmarcher, M., Rumetshofer, E., Clevert, D. A., Hochreiter, S., & Klambauer, G. (2019). Accurate prediction of biological assays with high-throughput microscopy images and convolutional networks. Journal of chemical information and modeling, 59(3), 1163-1171.
> >
> > [2] Palma, A., Theis, F. J., & Lotfollahi, M. (2023). Predicting cell morphological responses to perturbations using generative modeling. bioRxiv, 2023-07.
> >
> > [3] Bourou, A., Boyer, T., Gheisari, M., Daupin, K., Dubreuil, V., De Thonel, A., ... & Genovesio, A. (2024, October). PhenDiff: Revealing Subtle Phenotypes with Diffusion Models in Real Images. In International Conference on Medical Image Computing and Computer-Assisted Intervention (pp. 358-367). Cham: Springer Nature Switzerland.
> >
> > ---
> > **Regarding “Clarification on Novelty Claims”:**
> >
> > We appreciate your valuable suggestions to clarify and highlight the contributions of our work more explicitly. To the best of our knowledge, MorphoDiff is the first diffusion-based generative model that:
> > Enables high-resolution cellular phenotype prediction guided by perturbation signals.
> > Offers the flexibility to incorporate diverse perturbation embedding modules.
> > Is validated comprehensively on three public microscopy datasets using three evaluation approaches: (a) image fidelity metrics, (b) biological interpretability, and (c) visual assessment.
> >
> > In comparison, studies such as IMPA and PhenDiff have primarily focused on learning cellular patterns in image patches cropped around single cells, which differs in the scale and complexity of the generative task. While this approach is valuable for understanding cellular phenotypes, it does not leverage intercellular relationships in the whole well environment and limits the practical validation of the generated images in a well-based experimental setting.
> >
> > Moreover, in the PhenDiff framework, perturbation information is treated as a class label rather than as an informative embedding integrated into the generative process. This limitation prevents the model from generating images for unseen perturbations—whether genetic or small molecule-based. Their experiments were also limited to two datasets: (1) the BBBC021 dataset with three compounds and (2) fluorescent microscopy images of HeLa cells treated with DMSO and Nocodazole.
> >
> > In contrast, MorphoDiff addresses these gaps by integrating perturbation embeddings into the generative process, enabling generalization to unseen perturbations. While we acknowledge that its performance is more reliable for samples with embeddings similar to the training set, leaving room for further improvement, MorphoDiff supports a broader range of validation and application scenarios. We hope our work provides a strong foundation for developing advanced generative pipelines in this direction, contributing to progress in the drug discovery domain.
> > We hope the provided explanation clarifies our claim.
> >
> >
> > ---
> > **Regarding “Insufficient Detail in Key Methodological Components”:**
> >
> > Thank you for highlighting the importance of reproducibility in our work. All our experiments are based on publicly available models and datasets, with links provided in Appendix Note A.3.2 (Datasets section), along with detailed data processing descriptions. We have outlined the main steps required for reproducibility and included links to the pretrained checkpoints and scripts in the manuscript.
> >
> > We are committed to open science and will release the code for training, validation, and model checkpoints upon acceptance. We have experienced the challenges posed by insufficient resources and technical details when attempting to reproduce other tools. We emphasize that ensuring reproducibility in published works is essential for enabling robust benchmarking and improving the quality of scientific publications.
> >
> > For the reviewers, we will provide access to the code used for training and generating results. The code will be made publicly available upon acceptance.

---

> > > ### Author Response · Authors · 2024-11-23
> > >
> > > **Regarding “Overlooked Metrics for Sample Variability”:**
> > >
> > > The reviewer raises an excellent point regarding the diversity of generated images. To address this, we calculated the SSIM metric for 5,000 random pairs of cell-cropped patches within each compound, using the same dataset on which the additional benchmarking was performed. For MorphoDiff-generated images, the SSIM values were in the range of 0.2–0.3, which is similar to the range observed for ground truth images within each compound. Please find the averaged score for each compound below:
> > >
> > > AZ138 (GT: 0.27, MorphoDiff: 0.26),
> > >
> > > AZ258 (GT: 0.21, MorphoDiff, 0.16),
> > >
> > > Cytochalasin B (GT: 0.20, MorphoDiff: 0.15),
> > >
> > > Taxol (GT: 0.24, MorphoDiff: 0.22),
> > >
> > > Vincristine (GT: 0.30, MorphoDiff: 0.25)
> > >
> > > The low SSIM scores suggest that the generated images are not structurally similar, indicating variability. However, we acknowledge that SSIM primarily captures luminance, contrast, and structural similarity, and it may not fully reflect the desired cellular diversity for each compound. We hope this analysis addresses the reviewer’s concern and welcome any further suggestions for evaluating image diversity.
> > >
> > >
> > > ---
> > > **Regarding “Sensitivity to Diffusion Model Selection”:**
> > >
> > > While diffusion models have demonstrated remarkable capabilities in image generation, we agree that architectural choices can influence the quality of the generated images, particularly in the context of cellular phenotypes in our problem. To the best of our knowledge, there are few baseline models for large field-of-view microscopy image generation - Mol2Image and our work to the best of our knowledge. However, the PhenDiff paper employed another variation of the Diffusion models (Denoising Diffusion Implicit Models (DDIMs)) for class-conditional image generation in cropped cell patches. While the scope, complexity, and validation approach of their task differ from MorphoDiff, their provided results and sample images highlight the potential of diffusion model variants in this domain.
> > > We believe diffusion models hold significant promise for cellular morphology generation, paralleling their success in natural image generation as demonstrated by models like DALLE and Stable Diffusion. Although our current work did not focus on exploring alternative diffusion model architectures, further investigation into architectural choices and technical details could yield valuable insights and help identify optimal pipeline configurations for this critical application.

---

> > > ### Comment · Reviewer_qrds · 2024-11-26
> > >
> > > Thank you for your thorough response. I look forward to seeing your repository after the review period!

---

### Official Review · Reviewer_RSvH · 2024-10-30

**Soundness:** 4
**Presentation:** 3
**Contribution:** 4
**Rating:** 8
**Confidence:** 4

**Summary:**

The paper introduces MorphoDiff, a novel generative model designed to predict high-resolution cellular morphologies in response to different chemical and genetic perturbations. Using a latent diffusion model and a perturbation projection module, MorphoDiff generates images that mimic cellular phenotypes across multiple datasets. The model is validated through common generative modelling metrics, visual assessments, and biologically interpretable feature analysis, showing its ability to capture perturbation-specific and mechanism-of-action (MOA)-specific morphology. MorphoDiff aims to support drug discovery by enabling in silico exploration of large perturbational landscapes.

**Strengths:**

1. **Proposed Impact:**

The MorphoDiff framework’s potential to generate biologically meaningful cellular images based on perturbations could greatly impact drug discovery and cellular profiling. This innovation might streamline in silico analysis of large drug libraries and provide a baseline for future generative work in this domain.

2. **Writing and Accessibility:**

The paper is well-organised and clearly written, allowing readers from diverse backgrounds to follow the technical details of the diffusion model, perturbation embedding, and validation steps.

3. **Thorough Experimental Validation:**

The authors provide comprehensive computational, visual, and biological validations across multiple datasets, which demonstrates the versatility and robustness of MorphoDiff. The quantitative metrics, qualitative results, and the consistent comparative performance against Stable Diffusion enhance the reliability of their claims.

4. **Biological Insight:**

By clustering CellProfiler features, the authors demonstrate that the generated images capture perturbation-specific cellular morphology that aligns with real biological signals.

5. **Generality Across Chemical and Genetic Perturbations:**
MorphoDiff’s capacity to handle diverse perturbation types — both genetic and chemical — indicates potential applications across a wide range of experimental conditions, making it a versatile tool in cellular morphology prediction.

**Weaknesses:**

1. **Statistical Significance Lacking Clarity:**

The term "statistically significant" is used frequently (e.g., in results regarding FID and KID metrics) without detailing which statistical tests were applied or providing p-values. The methodology section would benefit from specifying these statistical tests to support the significance claims convincingly. Please could the authors specify which statistical tests were used for comparing FID and KID metrics between MorphoDiff and the baseline, and to provide p-values where applicable.

2. **Missing Citations for Established Findings:**

 Statements that reference existing biological knowledge, such as "MorphoDiff images revealed biological evidence that corroborates established findings," should include appropriate citations. This would anchor these statements in established literature. Please provide specific examples of relevant literature or established findings that support the claims about biological evidence.

3. **Limited comparison to other methods:**

 The authors compare MorphoDiff to Stable Diffusion, however, overlook other methods like PhenDiff. Although PhenDiff was proposed for cropped images, this can easily be extended. Could the authors extend their comparison to include PhenDiff? If not, could the authors discuss why this was left out of comparison?

4. **Minor:**

- There are occasional instances of incorrect citation formatting, which might distract readers or reduce clarity. For example, “...generative models in computer vision Rombach et al. (2022) and integrating state-of-the-art…”. Rombach et al. (2022) should appear in parentheses here. There are numerous other examples of this. Furthermore, citations like “Saha et al.” are missing the publication year, which should be updated to 2024​.
- The sentence “comparing standard-processed and cropped images in the BBBC021 experiments showed a slight improvement…”, should point to the results. I assume this is referring to Appendix Table 5.

**Questions:**

1. **Clarity on Clustering Technique:**

The term “KNN clustering” is ambiguous. It’s unclear if the authors mean k-means clustering or a different method. Given the context of CellProfiler feature clustering, clarifying this point would improve readability and understanding of the methodology.

2. **Assessing Combination Therapies:**

Combination therapies, where multiple drugs or genetic modifications are applied simultaneously, are increasingly important in personalised medicine and complex disease treatment strategies. Assessing cellular responses to these combinations is crucial, as interactions between therapies can lead to synergistic effects, enhanced efficacy, or reduced side effects that might not be observed with single agents alone. Given MorphoDiff's promising results with single perturbations, could the model be adapted or extended to predict morphology for combination therapies? Specifically, what modifications would be necessary to capture the interactions between therapies accurately, and are there unique challenges in validating these predictions with current datasets?

---

> ### Author Response · Authors · 2024-11-24
>
> We thank the reviewer for their kind remarks on the contributions of our work and have answered the raised concerns below.
>
> ---
> **Regarding “Statistical Significance Lacking Clarity”:**
>
> Thanks for your great suggestion. The original manuscript explained in Appendix Note (Validation section) that a t-test with p-value threshold below 0.05 was used to determine statistical significance. We more specifically clarified and updated that in the main text (Experiments - Comparison subsection) and Appendix (Note - Validation subsection) stating two-sample t-tests were used for statistically significance analysis with 0.05 threshold.
> To provide further details regarding the range of p-values, we added the following notations to tables: bold values indicating p-value < 0.05, * indicating p-value < 0.01, and ** indicating p-value < 0.001. Hope the revised manuscript provided enough details regarding performed analysis.
>
>
> ---
> **Regarding “Missing Citations for Established Findings”:**
>
> The mentioned sentence is at the beginning of a paragraph that aims to provide examples of biological evidence observed in the generated images that are consistent with the real images, such as the correlation of averaged CellProfiler features between the most and least similar compound treated images. To address the reviewer’s point and improve clarity, we revised the sentence to: "In addition to these major results, we observed that MorphoDiff images replicated certain biological nuances present in real images.", followed by explaining the results and their relation with real images. Thanks for your suggestion.
>
>
> ---
> **Regarding minor comments:**
>
> **minor comment #1:** We followed the citation guidelines provided in the official ICLR 2025 LaTeX template (https://github.com/ICLR/Master-Template/raw/master/iclr2025.zip), which state:
>
> "When the authors or the publication are included in the sentence, the citation should not be in parentheses, using \citet{} (e.g., 'See Hinton et al. (2006) for more information.'). Otherwise, the citation should be in parentheses using \citep{} (e.g., 'Deep learning shows promise to make progress towards AI (Bengio & LeCun, 2007).')."
>
> According to this guideline, citations included within a sentence should not appear in parentheses. However, we understand the reviewer’s concern about potential distractions and have moved some citations to the end of sentences where appropriate. Additionally, we corrected the year for the Saha et al. reference and ensured that all references include the correct year information.
>
>
> **minor comment #2:** Thank you for noticing reference to the Appendix Table 5 is missing. We have updated the manuscript to include the reference to the appendix table.
>
>
> ---
> **Regarding “Clarity on Clustering Technique”:**
>
> Thank you for catching this. The reference to KNN was a typo; the correct algorithm used in this evaluation is the K-Means clustering algorithm. We have corrected this in the revised manuscript.

---

> > ### Author Response · Authors · 2024-11-24
> >
> > **Regarding “Assessing Combination Therapies”:**
> >
> > Very interesting idea! The diverse applications of diffusion models in the computer vision domain have demonstrated their ability to incorporate multiple complex conditions into generative pipelines [1, 2, 3]. Building on the promising results of MorphoDiff, we agree that extending such models, including MorphoDiff, is both an interesting and necessary step toward enabling multi-condition morphology generation in more complex scenarios. These extensions include incorporating drug dosage values and exploring combinatorial therapies by combining multiple gene perturbations, or drugs and genes, to identify the most effective treatments.
> >
> > To achieve this, we propose two potential approaches:
> >
> > **Modular Design:** Treat the (combinatorial) perturbation encoder as a separate module and incorporate it into the MorphoDiff generative pipeline, which provides the flexibility of integrating with different tools.
> >
> > **Joint Learning:** Train the generative pipeline alongside the condition encoding module, enabling an integrated learning process.
> >
> > Developing foundational models capable of learning informative embeddings from multiple input covariates—such as perturbation information (dosage, chemical structure, and bioactivity data - for single or combined perturbations)—could significantly benefit various applications and phenotype prediction tasks influenced by diverse interventions.
> >
> > Regarding data availability, a recent review [4] highlighted examples of Cell Painting assays being used to elucidate the biological effects of various compound mixtures. However, studies providing phenotypic profiling for compound mixtures remain scarce, with only a few examples, such as [5] and [6]. Given the complexity of combinatorial therapy, large-scale datasets are essential. Training on such datasets would require substantial computational resources, including more GPUs and expanded storage capacities, to provide the model sufficient time and capability to learn intricate patterns effectively.
> >
> > While this remains a challenging task, we believe that research like ours underscores the potential of advanced generative models in cellular morphology generation. We hope this work inspires further efforts in this direction and encourages large technology companies to invest the necessary resources to tackle this critical and impactful challenge. We appreciate this suggestion and have now mentioned the potential for combination treatments to the conclusion.
> >
> > References:
> >
> > [1] Zhan, Z., Chen, D., Mei, J. P., Zhao, Z., Chen, J., Chen, C., ... & Wang, C. (2024). Conditional Image Synthesis with Diffusion Models: A Survey. arXiv preprint arXiv:2409.19365.
> >
> > [2] Nichol, A., Dhariwal, P., Ramesh, A., Shyam, P., Mishkin, P., McGrew, B., ... & Chen, M. (2021). Glide: Towards photorealistic image generation and editing with text-guided diffusion models. arXiv preprint arXiv:2112.10741.
> >
> > [3] Ramesh, A., Dhariwal, P., Nichol, A., Chu, C., & Chen, M. (2022). Hierarchical text-conditional image generation with clip latents. arXiv preprint arXiv:2204.06125, 1(2), 3.
> >
> > [4] Seal, S., Trapotsi, M. A., Spjuth, O., Singh, S., Carreras-Puigvert, J., Greene, N., ... & Carpenter, A. E. (2024). A Decade in a Systematic Review: The Evolution and Impact of Cell Painting. ArXiv.
> >
> > [5] Rietdijk, J., Aggarwal, T., Georgieva, P., Lapins, M., Carreras-Puigvert, J., & Spjuth, O. (2022). Morphological profiling of environmental chemicals enables efficient and untargeted exploration of combination effects. Science of the Total Environment, 832, 155058.
> >
> > [6] Pierozan, P., Kosnik, M., & Karlsson, O. (2023). High-content analysis shows synergistic effects of low perfluorooctanoic acid (PFOS) and perfluorooctane sulfonic acid (PFOA) mixture concentrations on human breast epithelial cell carcinogenesis. Environment International, 172, 107746.

---

> > > ### Comment · Reviewer_RSvH · 2024-11-25
> > > **Thank you!**
> > >
> > > I thoroughly appreciate this discussion. Thank you!

---

> > > > ### Author Response · Authors · 2024-11-25
> > > >
> > > > Thank you very much for your feedback and the clarification regarding the citations. We have updated the manuscript as suggested and would be happy to provide further clarification if any concerns remain.

---

> > > > > ### Comment · Reviewer_RSvH · 2024-12-01
> > > > >
> > > > > I appreciate the thorough work done by the authors to improve their manuscript. I believe that this is a great piece of work that should be commended. In particular, I believe that this work could have a large impact on in silico drug screening. I will raise my score to an 8.

---

> > > > > > ### Author Response · Authors · 2024-12-01
> > > > > >
> > > > > > We sincerely appreciate your helpful suggestions and thoughtful feedback on our manuscript, as well as your recognition of our work and the importance of the problem we aim to address.

---

> > ### Comment · Reviewer_RSvH · 2024-11-25
> > **Regarding in text citation**
> >
> > Thank you for your response. Regarding the point on in text citation, I still believe the authors have it wrong here in Line 76-78
> >
> > "By leveraging advanced generative models in computer vision Rombach et al. (2022) and integrating SOTA perturbation encoding modules, including single-cell foundational models Cui et al. (2024)".
> >
> > The problem here is that the citations are not integrated into the grammatical structure of the sentence. If the authors were to use \citet{} instead of \citep{} here, the sentence would need to be changes to something like:
> >
> > "By leveraging advanced generative models in computer vision, as demonstrated by Rombach et al. (2022), and integrating state-of-the-art perturbation encoding modules, such as single-cell foundational models proposed by Cui et al. (2024), we...".
> >
> > I hope that this makes sense.

---

> ### Author Response · Authors · 2024-11-24
>
> **Regarding “Limited comparison to other methods”:**
>
> We appreciate the reviewer’s suggestions to include benchmarking against other potential baselines and improve validation experiments. We fully recognize the importance of comparing our model to other published methods to provide a comprehensive evaluation of its capabilities, as well as the need for meaningful validation to motivate realistic applications. Below, we outline our efforts to address these concerns:
>
> 1. **Comparison with Mol2Image:**
>  We identified the Mol2Image paper as the most relevant to our pipeline in terms of input/output parameters and image scale. We tried our best to add this comparison to the experiments. The authors provided model weights for a dataset they used in their study [1]. However, no script or guideline are provided for training or fine-tuning to validate their method on other datasets. We also reached out multiple times to the authors directly but have not received any response and were unable to obtain the necessary material for adapting their model to our experiments and perform benchmarking.
>
> 2. **Comparison with IMPA and PhenDiff:**
>  We have successfully added benchmarking of MorphoDiff against the IMPA model [2] using their provided checkpoint for the BBBC021 dataset (trained on six conditions: five compounds and DMSO, provided in https://zenodo.org/record/8307629). Our models, trained on 14 compounds, were adapted for image quality evaluation. Using a pre-trained generalist cell segmentation model (https://github.com/Lee-Gihun/MEDIAR), we segmented generated and real images to obtain cell masks. Consistent 96×96 pixel patches centered around single cells (with IMPA images) were cropped for fair evaluation against IMPA images, and compound-specific FID and KID scores were calculated. Following the Comparison section in the main manuscript and to provide a robust evaluation, 500 cell patches were randomly selected for ground truth and the same amount were generated for each model (Stable Diffusion, IMPA, and MorphoDiff) across ten random seeds, with image quality metrics and significance tests reported for each compound as provided in the below table.
>
> | Model            | AZ138 (FID/KID) | AZ258 (FID/KID) | Taxol (FID/KID) | Cytochalasin B (FID/KID) | Vincristine (FID/KID) |
> |-------------------|-----------------|-----------------|-----------------|--------------------------|-----------------------|
> | Stable Diffusion | 1.40 / 0.14     | 0.94 / 0.08     | 1.50 / 0.15     | **0.83 / 0.07**          | 1.83 / 0.20          |
> | IMPA             | 0.98 / 0.07     | 1.18 / 0.12     | 1.28 / 0.11     | 1.23 / 0.11              | 1.05 / 0.07          |
> | MorphoDiff       | **0.82 / 0.06** | **0.76 / 0.05** | **1.09 / 0.10** | 1.19 / 0.11              | **0.86 / 0.06**      |
>
> Statistical testing (t-tests between the best and the second best method) demonstrated that MorphoDiff outperforms all other methods across all compounds, except for Cytochalasin B. Our investigation showed that high resolution images of Cytochalasin B treated images look more similar to real perturbed images for this drug compared to Stable Diffusion. The obtained FID and KID metrics in high resolution scale also showed MorphoDiff outperforms Stable Diffusion (both with standard preprocessing and cropped preprocessing), with sample images provided in the newly added Appendix Figure 6). We believe generating and validation of images with larger fields of view enables capturing cellular density and intercellular relationships, which will provide a more comprehensive insight of the cellular interactions and phenotypic shifts induced by different perturbations, as well as cellular diversity within the well. Detailed explanations of this analysis are provided in Appendix Note Validation subsection.
>
> For the PhenDiff model [3], we unfortunately encountered challenges in reproducing it due to the absence of available checkpoints and detailed guidelines for conditional data preparation. Despite reaching out to the authors, we have not received further information to facilitate reproducing their results.
>
> We hope this additional validation further demonstrates MorphoDiff's potential to advance virtual phenotypic screening tasks. Thank you for emphasizing this critical aspect, which has helped substantially enhance the strength of our paper.

---

> > ### Author Response · Authors · 2024-11-24
> >
> > References for the previous comment:
> >
> > [1] Hofmarcher, M., Rumetshofer, E., Clevert, D. A., Hochreiter, S., & Klambauer, G. (2019). Accurate prediction of biological assays with high-throughput microscopy images and convolutional networks. Journal of chemical information and modeling, 59(3), 1163-1171.
> >
> > [2] Palma, A., Theis, F. J., & Lotfollahi, M. (2023). Predicting cell morphological responses to perturbations using generative modeling. bioRxiv, 2023-07.
> >
> > [3] Bourou, A., Boyer, T., Gheisari, M., Daupin, K., Dubreuil, V., De Thonel, A., ... & Genovesio, A. (2024, October). PhenDiff: Revealing Subtle Phenotypes with Diffusion Models in Real Images. In International Conference on Medical Image Computing and Computer-Assisted Intervention (pp. 358-367). Cham: Springer Nature Switzerland.

---

### Official Review · Reviewer_EJPX · 2024-11-02

**Soundness:** 2
**Presentation:** 3
**Contribution:** 2
**Rating:** 3
**Confidence:** 4

**Summary:**

The paper addresses the challenge of virtual phenotypic screening for drug discovery.  The authors train a guided 2D latent diffusion model on microscopy images (with Cell Painting markers) to generate images of cells, as they would appear after treatment with a genetic or chemical perturbation.  In order to guide the model, they use embeddings of the corresponding perturbation: RDKit for compounds and scGPT for genes.

The authors fine-tune the publicly available Stable Diffusion v1.4 on three different Cell Painting datasets: Rxrx1 (siRNA), BBC021 (compound) and Rohban (over-expression), and compare the impact of the Perturbation Identifier Module for guiding the image generation process.  In addition to standard image quality metrics (FID and KID), the authors also use the popular Cell Profiler framework to extract engineered features from both real and generated images, and compare their ability to cluster perturbations by perturbation ID (ie, replicate) and mechanism of action (MOA).  The authors also evaluate correlation of Cell Profiler features extracted from real and generated images, demonstrating the impact of guiding the image generation process with perturbation embeddings.

**Strengths:**

The paper is clearly described, including a thorough description of the model, datasets and training procedure.  It also includes links to code for data conversion, the pre-trained Stable Diffusion model, and author's fine-tuning script, which taken together provide a high level of reproducibility.

One of the main contributions of the paper is to apply state-of-the-art guided diffusion models to the task of virtual phenotypic screening.  Other papers have similarly applied generative models (GANs and flow-based models); however, this is the first application of diffusion models (that I'm aware of) to this task.  The generated images look high quality, despite fine-tuning on a relatively small training set of Cell Painting images.

In addition, the authors evaluate their method on 3 independent open-source datasets: Rxrx1, BBBC021 and Rohban.

**Weaknesses:**

Major concerns:

My primary concern with the paper is regarding benchmarking of the MorphoDiff model against other methods.  In particular, the only comparison that is included throughout the paper is a baseline of Stable Diffusion (without guidance by the Perturbation Identifier module).  While this comparison is positive in all cases, in my view it is equivalent to an ablation study, demonstrating that the model is leveraging the information in the perturbation embeddings.  It doesn't demonstrate if the model accuracy is sufficiently high to be useful in virtual phenotypic screening, or compare the model performance against other published models such as Mol2Image, IMPA, or GRAPE (https://arxiv.org/pdf/2406.07763).  I suggest including additional discussion and comparisons of other published methods, and expanding the analysis of MOA inference to a larger set of MOA classes, as it's a core evaluation metric for phenotypic screening.

In addition, while the method is nicely implemented and clearly explained in the paper, I find it does not have high novelty compared to other methods (Mol2Image, IMPA, GRAPE) for generating images of cellular morphology under perturbation, and therefore a rigorous comparison of MorphoDiff's performance relative to the existing state-of-the-art would be very important to demonstrate a significant contribution from the paper.

Additional concerns:

The MorphoDiff model was fine-tuned on very small datasets, each on the order of 1000 perturbations.  Much larger Cell Painting datasets have been released in recent years, such as JUMP (120,000 perturbations) and Rxrx3 (20,000 perturbations).  Demonstrating the capability of MorphoDiff on these datasets would make it more impactful related to the current state-of-the-art for virtual phenotypic screening, such as MoCoP (https://www.biorxiv.org/content/10.1101/2023.05.01.538999v3.full.pdf).  I suggest the authors discuss the potential challenges of applying diffusion models to large Cell Painting datasets, and how these could be overcome to advance the field of generative virtual phenotypic screening.

The images generated based on perturbations seen during training look quite good.  However, in each case, they are shown for very strong drug treatments, such as cytoskeletal destabilizers.  It would be informative to demonstrate if MorphoDiff can capture more subtle phenotypes that are typically present in phenotypic screens.

In the conclusion, the authors suggest that future work could establish standardized evaluation frameworks for this emerging field.
 I recommend to check out the following paper that provides a framework and numerous metrics for evaluating perturbative maps: https://journals.plos.org/ploscompbiol/article?id=10.1371/journal.pcbi.1012463

In Figure 3b, very few MOA clusters were considered.  I suggest leveraging datasets with a larger number of available MOAs / gene clusters.  One such option would be to show the 1108 siRNA perturbations in Rxrx1 (used in the paper); however, due to the large off-target effects of siRNA that may not be very meaningful.  In which case, JUMP would provide another option.

In Figure 3c, comparing NMI of morphoDiff on Perturbation ID doesn't seem meaningful.  I may have misunderstood this point, but it seems that the variation in Cell Profiler features extracted from morphoDiff with a constant Perturbation ID should reflect only the noise in the generation process.  Whereas the real data will have large technical effects between replicates.  As such, I would expect a generative model to be more self-consistent than the raw data (which is known to be very noisy at the replicate level).  I suggest focusing evaluation on the MOA clustering (ideally on an expanded MOA set), and perhaps investigating the effect of diversity/noise in the generation process on the accuracy of MOA clustering.

In Figure 4a, the correlation coefficient between CellProfiler features extracted from MorphoDiff vs features extracted from real images for the same perturbation is close to zero.  I suggest including a separate measure of the correlation of CellProfiler features between replicates (real images) of the same perturbation, to provide a rough upper-limit for how strongly we could expect the model to correlate with real data.  If the model correlation is similar to the replicate correlation, then it can be argued more meaningfully that the model generated images are "interchangeable" with real data.

Finally, the authors observe that the similarity of images generated by MorphoDiff relies on the similarity of the output from the perturbation embedding module.  In other words, the model does not learn an improved perturbation embedding from the data, as for example Mol2Image and GRAPE do. As such, it's not clear to me that this approach would be informative in a virtual screening setting, where the goal is to learn a predictive embedding of unseen perturbations that improves generalization beyond eg, the performance of RDKit.  I suggest additional discussion on the implications of this limitation for virtual screening, and how they might be addressed through modifications to the Perturbation Embedding Module.

**Questions:**

Suggestions (summarized from above):
1. Train the MorphoDiff model on larger CellPainting datasets, to improve the learned relationship between perturbation and phenotype
2. Evaluate model performance in the harder case of more subtle phenotypes
3. Conduct additional evaluations on datasets with a larger number of MOA clusters, leveraging the evaluation framework from: https://journals.plos.org/ploscompbiol/article?id=10.1371/journal.pcbi.1012463
4. Explore the possibility of updating the Perturbation Identifier Module with learned perturbation embeddings, which would provide a more useful output in the scenario of virtual phenotypic screening.

---

> ### Author Response · Authors · 2024-11-24
>
> We thank the reviewer for detailed review and great suggestions to improve our work. Please find our answers to the raise points below. We hope the added benchmarking and provided explanation address the reviewer's concerns.
>
> ---
> **Response to the reviewer’s concern regarding benchmarking:**
>
> We sincerely thank the reviewer for their constructive suggestions regarding the benchmarking and validation of MorphoDiff. We fully recognize the importance of comparing our model to other published methods to provide a comprehensive evaluation of its capabilities, as well as the need for meaningful validation to motivate realistic applications. Below, we outline our efforts to address these concerns:
>
> 1. **Comparison with Mol2Image:**
>   We identified the Mol2Image paper as the most relevant to our pipeline in terms of input/output parameters and image scale. We tried our best to add this comparison to the experiments. The authors provided model weights for a dataset they used in their study [1]. However, no script or guideline are provided for training or fine-tuning to validate their method on other datasets. We also reached out multiple times to the authors directly but have not received any response and were unable to obtain the necessary material for adapting their model to our experiments and perform benchmarking.
>
> 2. **Comparison with GRAPE:**
>    The GRAPE paper employed a style-transfer approach to learn gene-level feature representations from images of genetically perturbed cells. While relevant, the paper does not provide publicly available code, which posed a major obstacle for comparison [2].
>
> 3. **Comparison with IMPA and PhenDiff:**
>    We have successfully added benchmarking of MorphoDiff against the IMPA model [3] using their provided checkpoint for the BBBC021 dataset (trained on six conditions: five compounds and DMSO, provided in https://zenodo.org/record/8307629). Our models, trained on 14 compounds, were adapted for image quality evaluation. Using a pre-trained generalist cell segmentation model (https://github.com/Lee-Gihun/MEDIAR), we segmented generated and real images to obtain cell masks. Consistent 96×96 pixel patches centered around single cells were cropped for fair evaluation against IMPA images, and compound-specific FID and KID scores were calculated. Following the Comparison section in the main manuscript and to provide a robust evaluation, 500 cell patches were randomly selected for ground truth and the same amount were generated for each model (Stable Diffusion, IMPA, and MorphoDiff) across ten random seeds, with image quality metrics and significance tests reported for each compound as provided in the below table.
>
>
> | Model            | AZ138 (FID/KID) | AZ258 (FID/KID) | Taxol (FID/KID) | Cytochalasin B (FID/KID) | Vincristine (FID/KID) |
> |-------------------|-----------------|-----------------|-----------------|--------------------------|-----------------------|
> | Stable Diffusion | 1.40 / 0.14     | 0.94 / 0.08     | 1.50 / 0.15     | **0.83 / 0.07**          | 1.83 / 0.20          |
> | IMPA             | 0.98 / 0.07     | 1.18 / 0.12     | 1.28 / 0.11     | 1.23 / 0.11              | 1.05 / 0.07          |
> | MorphoDiff       | **0.82 / 0.06** | **0.76 / 0.05** | **1.09 / 0.10** | 1.19 / 0.11              | **0.86 / 0.06**      |
>
>
> Statistical testing (t-tests between the best and the second best method) demonstrated that MorphoDiff outperforms all other methods across all compounds, except for Cytochalasin B. Our investigation showed that high resolution images of Cytochalasin B treated images look more similar to real perturbed images for this drug compared to Stable Diffusion. The obtained FID and KID metrics in high resolution scale also showed MorphoDiff outperforms Stable Diffusion (both with standard preprocessing and cropped preprocessing), with sample images provided in the newly added Appendix Figure 6. We believe generating and validation of images with larger fields of view enables capturing cellular density and intercellular relationships, which will provide a more comprehensive insight of the cellular interactions and phenotypic shifts induced by different perturbations, as well as cellular diversity within the well. Detailed explanations of this analysis are provided in Appendix Note Validation subsection.
>
> For the PhenDiff model [4], we unfortunately encountered challenges in reproducing it due to the absence of available checkpoints and detailed guidelines for conditional data preparation. Despite reaching out to the authors, we have not received further information to facilitate reproducing their results.
>
> We hope this additional validation further demonstrates MorphoDiff's potential to advance virtual phenotypic screening tasks. Thank you for emphasizing this critical aspect, which has helped substantially enhance the strength of our paper.

---

> > ### Author Response · Authors · 2024-11-24
> >
> > **Continuing from the previous comment:**
> >
> > We also acknowledge that validation on larger datasets with a wider range of MOAs would enhance the robustness of the model’s validation. To the best of our knowledge, only the BBBC021 dataset provided this annotation among the processed datasets. Further discussion on the challenges associated with modeling and validating larger datasets is provided in our response to the next response.
> >
> > Reference:
> >
> > [1] Hofmarcher, M., Rumetshofer, E., Clevert, D. A., Hochreiter, S., & Klambauer, G. (2019). Accurate prediction of biological assays with high-throughput microscopy images and convolutional networks. Journal of chemical information and modeling, 59(3), 1163-1171.
> >
> > [2] Bigverdi, M., Höckendorf, B., Yao, H., Hanslovsky, P., Lopez, R., & Richmond, D. (2024). Gene-level representation learning via interventional style transfer in optical pooled screening. In Proceedings of the IEEE/CVF Conference on Computer Vision and Pattern Recognition (pp. 7921-7931).
> >
> > [3] Palma, A., Theis, F. J., & Lotfollahi, M. (2023). Predicting cell morphological responses to perturbations using generative modeling. bioRxiv, 2023-07.
> >
> > [4] Bourou, A., Boyer, T., Gheisari, M., Daupin, K., Dubreuil, V., De Thonel, A., ... & Genovesio, A. (2024, October). PhenDiff: Revealing Subtle Phenotypes with Diffusion Models in Real Images. In International Conference on Medical Image Computing and Computer-Assisted Intervention (pp. 358-367). Cham: Springer Nature Switzerland.
> >
> >
> > ---
> > **Regarding MorphoDiff’s novelty:**
> >
> > We appreciate the reviewer’s comment regarding the contributions of our work and the importance of demonstrating novelty and significance compared to existing studies.
> >
> > To the best of our knowledge, MorphoDiff is the first diffusion-based generative framework that:
> >
> > (1) Enables high-resolution cellular phenotype prediction guided by perturbation signals.
> >
> > (2) Offers the flexibility to incorporate diverse perturbation embedding modules.
> >
> > (3) Is validated comprehensively on three public microscopy datasets using three evaluation approaches: (a) image fidelity metrics, (b) biological interpretability, and (c) visual assessment.
> >
> > Existing studies, such as IMPA and PhenDiff have primarily focused on learning cellular patterns in tiny image patches cropped around single cells, which differs in the scale and complexity of the generative task. While this approach is valuable for understanding cellular phenotypes, it does not leverage intercellular relationships in the whole well environment and limits the practical validation of the generated images in a well-based experimental setting.
> >
> > Moreover, in the PhenDiff framework, perturbation information is treated as a class label rather than as an informative embedding integrated into the generative process. This limitation prevents the model from generating images for unseen perturbations—whether genetic or small molecule-based. Their experiments were also limited to two datasets: (1) the BBBC021 dataset with three compounds and (2) fluorescent microscopy images of HeLa cells treated with DMSO and Nocodazole.
> >
> > In contrast, MorphoDiff addresses these gaps by integrating perturbation embeddings into the generative process, enabling generalization to unseen perturbations. While we acknowledge that its performance is more reliable for samples with embeddings similar to the training set, leaving room for further improvement, MorphoDiff supports a broader range of validation and application scenarios. We believe our work sets a strong foundation for developing advanced generative pipelines in this direction.

---

> ### Author Response · Authors · 2024-11-24
>
> **Response regarding analysis of large datasets and MOA set:**
>
> We thank the reviewer for the thoughtful feedback and suggestions for enhancing the impact of our work. Indeed, recent releases of large-scale Cell Painting datasets, such as JUMP and RxRx3, represent valuable resources for developing machine learning tools that thrive on diverse and extensive data. These datasets hold immense potential for advancing the field of virtual phenotypic screening.
>
> However, as highlighted in [1], the primary challenges of leveraging these datasets are the substantial resources required for data handling, storage, and computation. In our work, we leveraged the existing datasets that fit our computational resources for training/validation, and storage resources for raw, preprocessed, and generated images across multiple experiments. Based on our estimation training all the models on the large Cell Paintings datasets could be prohibitively expensive. For example, it would take at least 100 GPU years for the JUMP [2] dataset to finish all the experiments based on our A40 GPU.
>
> Despite these challenges, we believe our work demonstrates the promise of diffusion-based models for cellular phenotype prediction and establishes a strong baseline for future research in this direction. Expanding MorphoDiff to larger datasets like RxRx3 and JUMP and more MOA labels would undoubtedly enhance its impact, but doing so would require access to significantly greater computational and storage resources.
>
> We hope our work inspires larger institutions and organizations with access to such resources to invest in this important area, leveraging these comprehensive datasets to further advance the capabilities of generative models in virtual phenotypic screening. We trust that the challenges we faced during the development and analysis of MorphoDiff are understandable and highlight the need for broader collaboration and resource allocation to tackle these exciting yet demanding problems.
>
> Reference:
>
> [1] Seal, S., Trapotsi, M. A., Spjuth, O., Singh, S., Carreras-Puigvert, J., Greene, N., ... & Carpenter, A. E. (2024). A Decade in a Systematic Review: The Evolution and Impact of Cell Painting. ArXiv.
>
> [2] Chandrasekaran, S. N., Ackerman, J., Alix, E., Ando, D. M., Arevalo, J., Bennion, M., ... & Carpenter, A. E. (2023). JUMP Cell Painting dataset: morphological impact of 136,000 chemical and genetic perturbations. BioRxiv, 2023-03.
>
>
> ---
> **Regarding capturing subtle phenotype:**
>
> We appreciate the reviewer’s valuable suggestion. In alignment with this, we have analyzed datasets with varying types of perturbations (genetic and chemical) and levels of complexity in our experiments. Our validation using perturbation-specific image distance metrics demonstrated meaningful improvements in capturing perturbation-specific signals across different datasets and different perturbation subsets, including those with subtle and more visually detectable phenotypes (such as the aurora kinase inhibitor AZ258 or the SFK inhibitor PP2).
>
> Moreover, MorphoDiff was able to capture relatively rare cellular phenotypes which are not subtle in their morphology but in their frequency such as the small subset of metaphase-arrested cells induced by Colchicine, AZ841, and AZ258 in real images (sample images provided in the newly added Appendix Figure 7). Similar smaller, round, and bright structures were also identified in the corresponding generated cells, although the level of detail is less refined compared to the real images. We believe this is because metaphase-stage cells are too rare in the real dataset. However, the model successfully captures some features of these rare cellular events.
>
>
> ---
> **Regarding conclusion:**
>
> Thank you for sharing this recent paper—it is a great resource that proposes unified benchmarking steps and standards across different modalities, including imaging and RNA sequencing data, with a focus on validating sample embeddings and their biological relationships.
> As well, to clarify, for the image generation task, we believe alongside biological validation (which is the primary focus of the shared paper), additional criteria should also be considered, such as pixel distribution distance which is common in evaluation of generative models. We agree the mentioned paper mostly aligns with our proposal (more specifically in the biological validation context), and have updated the conclusion section to reflect more specific future directions of this research domain.
>
> ---
> **Regarding adding upper-limit to Figure 4a:**
>
> Great idea to use the correlation between real images as an upper bound. We improved Figure 4 a by adding an upper bound value by calculating the correlation between random split of real CellProfiler features of each condition.

---

> > ### Author Response · Authors · 2024-11-24
> >
> > **Regarding NMI interpretation:**
> >
> > We thank the reviewer for the detailed feedback and suggestions to improve and clarify this analysis.
> > As noted, real microscopy images inherently contain batch effects, such as plate or well effects, which contribute to technical variability. To mitigate (but not eliminate) this, we applied illumination correction following the approach outlined in the RxRx1 paper [1], as described in the Appendix Note Pre-Processing subsection.
> >
> > Moreover, both real and generated images exhibit diversity in cellular morphology influenced by the dosage of compound treatments. As highlighted in the BBBC021 paper [2]: “The phenotypic response of cells following chemical perturbations is inherently unstable; multiple phenotypes may appear transiently with distinct temporal dynamics for any specific compound and dose.” This inherent variability in the BBBC021 dataset results from compound treatments at multiple doses, leading to a range of phenotypes within the same compound. Some phenotypes may closely resemble DMSO-treated cells or other compounds, which contributes to lower NMI scores for compound/MOA-based clustering in real images.
> >
> > For the generated images, your observation is valid—condition-specific generations capture compound-specific signals and have improved the clustering performance compared to real images (Figure 3 c). However, they also reflect the learned diversity in image patterns, mirroring the variability seen in real samples. This diversity includes a range of cellular morphologies, some of which overlap between different compounds. Such overlap can challenge clustering algorithms, reducing the NMI scores for grouping images by ground-truth compound or MOA labels.
> >
> > We agree that validation on MOA clustering, potentially with an expanded MOA set, could provide additional insights into the model's performance (we have discussed this in response to comment #4). Additionally, investigating the impact of diversity or noise in the generation process on MOA clustering accuracy is an interesting direction and aligns well with your suggestion.
> > Thank you again for your thoughtful comments and for highlighting these important points. We will incorporate these insights into future analyses.
> >
> > References:
> >
> > [1] Sypetkowski, M., Rezanejad, M., Saberian, S., Kraus, O., Urbanik, J., Taylor, J., ... & Earnshaw, B. (2023). Rxrx1: A dataset for evaluating experimental batch correction methods. In Proceedings of the IEEE/CVF Conference on Computer Vision and Pattern Recognition (pp. 4285-4294).
> >
> > [2] Caie, P. D., Walls, R. E., Ingleston-Orme, A., Daya, S., Houslay, T., Eagle, R., ... & Carragher, N. O. (2010). High-content phenotypic profiling of drug response signatures across distinct cancer cells. Molecular cancer therapeutics, 9(6), 1913-1926.
> >
> >
> > ---
> > **Regarding Perturbation Embedding Module:**
> >
> > The reviewer's interpretation is correct—part of the generative model's performance relies on the informativeness of the perturbation embeddings. While this modular approach offers the flexibility to use various tools for perturbation encoding (as this remains an active area of research [1, 2, 3]), it also introduces the limitation of the model’s dependence on the quality of these embeddings.
> >
> > We appreciate the reviewer’s insightful suggestion and have updated the discussion in the “MorphoDiff generalizability to unseen drugs” subsection of the manuscript to highlight the potential extension of MorphoDiff with joint learning of conditions.
> >
> > Reference:
> >
> > [1] Kim, H., Lee, J., Ahn, S., & Lee, J. R. (2021). A merged molecular representation learning for molecular properties prediction with a web-based service. Scientific Reports, 11(1), 11028.
> >
> > [2] Blay, V., Radivojevic, T., Allen, J. E., Hudson, C. M., & Garcia Martin, H. (2022). MACAW: an accessible tool for molecular embedding and inverse molecular design. Journal of chemical information and modeling, 62(15), 3551-3564.
> >
> > [3] Li, T., Huls, N. J., Lu, S., & Hou, P. (2024). Unsupervised manifold embedding to encode molecular quantum information for supervised learning of chemical data. Communications Chemistry, 7(1), 133.

---

> ### Comment · Reviewer_EJPX · 2024-11-27
>
> Regarding larger datasets with wider range of MOAs:
>
> > We also acknowledge that validation on larger datasets with a wider range of MOAs would enhance the robustness of the model’s validation. To the best of our knowledge, only the BBBC021 dataset provided this annotation among the processed datasets.
>
> MOAs can be extracted from sources such as ChEMBL for many compounds in public screening datasets.  I encourage the authors to check out the following resource for additional ground truth and evaluation metrics that can be leveraged in future studies: https://journals.plos.org/ploscompbiol/article?id=10.1371/journal.pcbi.1012463
>
> > Based on our estimation training all the models on the large Cell Paintings datasets could be prohibitively expensive. For example, it would take at least 100 GPU years for the JUMP [2] dataset to finish all the experiments based on our A40 GPU.
>
> The computational challenges of scaling diffusion models to larger datasets is a clear challenge that would need to be overcome before this method can be deployed at scale.  While weakly supervised and contrastive learning models have been deployed on the full JUMP datasets, it still remains to be seen whether the benefits of latent diffusion models outweigh the computational challenges of scaling them up to JUMP-sized datasets.

---

> > ### Author Response · Authors · 2024-11-27
> >
> > We greatly appreciate the reviewer’s feedback on our responses and will consider the suggestions in future studies (with the shared paper already referenced in the introduction section).
> >
> > We hope our current study inspires further exploration of the advantages and limitations of diffusion models (and generative models in general) in larger scale for this application.

---

### Official Review · Reviewer_Pgmp · 2024-11-09

**Soundness:** 3
**Presentation:** 3
**Contribution:** 3
**Rating:** 8
**Confidence:** 3

**Summary:**

This paper presents a novel generative model framework called MorphpDiff, which produces high-resolution cell morphology images in response to a drug or genetic perturbations. The authors leverage the use of latent diffusion models combined with perturbation-specific embeddings to simulate cellular response.  This model provides an alternative to traditional lab assays/screens by generating silicon drug and genetic responses in the cell. MorphoDiff is validated using three publicly available datasets (RxRx1, BBBC021, and Rohban et al. l Cell painting dataset)  ensuring that they have cell shape and state variability and perturbations. These models are evaluated with common image fidelity and metrics including Frechet Inception Distance (FID) and kernel Inception Distance (KID) showing improvements over the standard stable diffusion. Most importantly, the authors were able to capture cell morphology changes using CelProfiler for PCA and other morphology-based metrics.

**Strengths:**

MorphoDiff method shows novel methods combining latent diffusion models with perturbation embedding to predict cellular morphologies. The model generalizes across different cell types, drugs, and genetic alterations better than the standard stable diffusion model. It performs better than the standard diffusion model showing that the perturbation embedding helps the model learn and has the dual capability with gene knock-in/out and small molecule perturbations.
The authors use public and diverse datasets to validate their model and have clear reporting on how these were used. The model seems to generalize across different perturbations. The authors use generative metrics as well as biological meaningful metrics to evaluate the performance of their model. The models are tested using tools like CellProfiler to get interpretable and comparable results to the ground truth biological data.

This method scales and enables larger high-content screens for drugs. It provides a tool for simulating the effects of drugs or other genetic perturbations.

**Weaknesses:**

MorphoDiff fails to generalize to compounds that are out-of-distribution, and seems like it performs better at the lower frequency than the high-frequency features. How accurate is this model and how consistent is it at capturing other cellular structures across the cell?
In many of these stages, cell cycle plays a big role. Here, the paper lacks exploring how their model handles more complex events in the fields of view such as cell division.
This method relies on RGB images, it would be interesting to do an ablation study and see the performance with fewer channels.

**Questions:**

- Could the authors clarify the failure modes of MorphoDiff and the range of validity? How does the model perform across a broad range of drugs and genetic perturbations? Is there a category of drugs where the performance drops?

- What kind of data augmentations were made and how do these impact the generalization across different drugs and genetic perturbations? I suggest ablations to improve the model's robustness. Additionally, could the authors consider conducting ablation studies to identify which augmentations make the model more robust?

- What are the other factors were included in the selection of perturbation conditions? Did you add prior information such as concentrations of chemicals?

---

> ### Author Response · Authors · 2024-11-23
>
> We thank the reviewer for their kind remarks on the contributions of our work and providing great suggestions. Below are our answers to your questions.
>
> **Regarding the generalizability of MorphoDiff in out-of-distribution settings:**
>
> Thank you to the reviewer for highlighting the importance of evaluating MorphoDiff in out-of-distribution (OOD) scenarios. Given the vast space of unseen perturbations during training and the inherent complexity of OOD conditional image generation, we looked into the correlation between embeddings of OOD compounds in the BBBC021 experiment and those of in-distribution compounds. Our validation focused on the top 15 OOD compounds with correlation values ranging from 0.46 to 0.92, as detailed in Appendix Table 7, which showed statistically significant improvements in image distance metrics (t-test, p-value < 0.05). Sample images of the top five compounds (correlation range: 0.61–0.92) are provided in Figure 4b.
> While these results demonstrate promising performance for unseen compounds with correlations above 0.46, we acknowledge more extensive experiments would provide better insight into the model's reliability in OOD scenarios, and we will take this suggestion into account by involving more diverse and complex settings in future extended work. We appreciate the reviewer's feedback.
>
> ---
>  **Regarding MorphoDiff’s consistency in capturing cellular structures across the cells:**
>
> To evaluate MorphoDiff’s accuracy and consistency in capturing cellular biological features, particularly cellular structure, we leveraged the well-established CellProfiler tool. We analyzed multiple feature subsets describing various cellular aspects, including primary compartments such as the cell, nucleus, and cytoplasm. We examined feature sets reflecting cellular structure through elements of shape (AreaShape), texture (Texture), and a selection of Zernike polynomial-based features to capture cellular morphology.
> Our results, shown in Figure 3(d), Figure 4 (a), Appendix Figures 11, 12, and 13, demonstrated that across different compounds and processing methods, MorphoDiff-generated images shifted CellProfiler feature space towards the feature distribution of real perturbed cells and outperformed the baseline. Although a wide range of CellProfiler feature subsets could be explored depending on research goals and data characteristics, we selected these sets based on their biological relevance representing various cellular properties. We hope the provided explanation addresses the reviewer’s concerns; however, we would be glad to clarify further with additional specifics if needed.
>
> ---
> **In many of these stages, cell cycle plays a big role. Here, the paper lacks exploring how their model handles more complex events in the fields of view such as cell division:**
>
> Thank you for highlighting this important aspect of the generated images. Together with our expert collaborators, we analyzed the generated samples for each compound and observed examples of cells in various states beyond typical interphase cells. Capturing cell division across most classes in the Cell Painting dataset is challenging as the majority of cells are in the interphase stage (not dividing), as clearly illustrated in Figures 2, Figure 4, and Appendix Figure 8. Drug cytochalasin B, which inhibits cytoplasmic division by blocking the formation of contractile microfilaments, results in a higher proportion of bi-nucleated cells [1]. We specifically examined real and generated images with this drug and identified several instances of bi-nucleation in the generated samples as the newly added Appendix Figure 7 demonstrates, indicating the capacity of the model to capture the cell cycle events. Additionally, we observed that under certain drug treatments, such as Colchicine, AZ841, and AZ258, which arrest cells in metaphase, a small subset of cells remains in the metaphase stage in the real images, as shown in the Appendix Figure 7. Similar smaller, round, and bright structures were also identified in the generated cells, although the level of detail is less refined compared to the real images. We believe this is because metaphase-stage cells are too rare in the real dataset. However, the model successfully captures some features of these rare cellular events.
> We hope the provided explanations and examples address the reviewer’s concern.
>
> Reference:
>
> [1] Theodoropoulos, P. A., Gravanis, A., Tsapara, A., Margioris, A. N., Papadogiorgaki, E., Galanopoulos, V., & Stournaras, C. (1994). Cytochalasin B may shorten actin filaments by a mechanism independent of barbed end capping. Biochemical pharmacology, 47(10), 1875-1881.

---

> > ### Author Response · Authors · 2024-11-23
> >
> > **Regarding ablation study:**
> >
> > This is an interesting experiment and has been investigated in previous studies showing minimal effect of dropping measurement of any channel from the Cell Painting panel in their performance validation [1][2].
> >
> > The optimal choice of channels for analyzing a dataset depends on its specific characteristics (e.g., cell lines, perturbations) and the research objectives. In our work, MorphoDiff was designed to take three channels as input. The selection of channels was made with expert guidance, considering perturbation type and the compartments that are interpretable and required for biological feature extraction (by CellProfiler in our work).
> > However, we think investigating the impact of both the number of channels and different channel combinations could be an exciting research direction in generative tasks. Active research in this area, such as [3], underscores the importance of this line of inquiry and its potential to enhance our understanding of how channel selection influences image generation and model performance.
> >
> > Reference:
> >
> > [1] Cimini, B. A., Chandrasekaran, S. N., Kost-Alimova, M., Miller, L., Goodale, A., Fritchman, B., ... & Carpenter, A. E. (2023). Optimizing the Cell Painting assay for image-based profiling. Nature protocols, 18(7), 1981-2013.
> > [2] Tromans‐Coia, C., Jamali, N., Abbasi, H. S., Giuliano, K. A., Hagimoto, M., Jan, K., ... & Cimini, B. A. (2023). Assessing the performance of the Cell Painting assay across different imaging systems. Cytometry Part A, 103(11), 915-926.
> > [3] Chen, Z. S., Pham, C., Wang, S., Doron, M., Moshkov, N., Plummer, B., & Caicedo, J. C. (2024). CHAMMI: A benchmark for channel-adaptive models in microscopy imaging. Advances in Neural Information Processing Systems, 36.
> >
> >
> > ---
> > **Could the authors clarify the failure modes of MorphoDiff and the range of validity? How does the model perform across a broad range of drugs and genetic perturbations? Is there a category of drugs where the performance drops?**
> >
> > Our observations from both qualitative and quantitative results indicate that the number of conditions and the phenotypic distinctiveness of treated cells impact the model's learning. When there are perturbation-specific signatures in cell morphology, the model is better able to capture these patterns and when the number of perturbations scales the task becomes more challenging. Moreover, extended training of the models and more training samples improve models' learning in capturing subtle cellular changes. We accounted for this parameter by conducting experiments on three diverse datasets, encompassing a range of perturbations from subtle to more pronounced phenotypes.
> >
> > Additionally, given that drug dosage is not incorporated into the current setting of generative process, it prevented us from performing dosage specific benchmarking on drugs that exhibit more distinct phenotypes for different drug concentrations. For a fair comparison and accounting the diversity of real images, we compared the generated images by the model with all training images from different dosages, when provided.
> > We believe that dedicating sufficient resources to train diffusion models on cellular morphology images (based on their promising performance in other domains), particularly within a scaled perturbational space, holds significant potential in capturing more nuanced cellular complexities. We hope our work provides a strong baseline and motivates researchers and institutes to invest more in this direction.
> > We appreciate the reviewer’s nudge to address these issues and have added a brief mention of them in the conclusion section

---

> > > ### Author Response · Authors · 2024-11-23
> > >
> > > **What kind of data augmentations were made and how do these impact the generalization across different drugs and genetic perturbations? I suggest ablations to improve the model's robustness. Additionally, could the authors consider conducting ablation studies to identify which augmentations make the model more robust?**
> > >
> > > We applied data augmentation to equalize sample sizes across perturbation groups during training to address the imbalance in the number of perturbed images across different conditions (e.g., Taxol-treated images in the BBBC021 dataset were significantly more abundant than other compounds). Specifically, we used random flipping and rotation, common deep learning techniques, to increase the number of training images for under-represented perturbations, while preventing the model from learning confounding factors like absolute cell orientation and position as relevant informative features. This augmentation strategy helped improve model generalizability by exposing the model to more balanced and diverse representations of each condition.
> > >
> > > Data imbalance can bias models towards over-represented conditions, and balancing techniques such as those used in our study have been employed in many other works to mitigate this issue. We applied these augmentations early in the training process and anticipate that removing them would reduce the model’s generalizability. Conducting ablation studies to explore the impact of augmentation on robustness is an excellent suggestion, and we appreciate the reviewer’s recommendation for future work. Unfortunately, given the timeline required for training diffusion models, this does not fit the deadline for the rebuttal.
> > >
> > > ---
> > > **What other factors were included in the selection of perturbation conditions? Did you add prior information such as concentrations of chemicals?**
> > >
> > > The selection of perturbation conditions in our experiments was guided by biological relevance. We referenced the original publications for each dataset to identify biologically meaningful perturbations for both modeling and validation. Specifically, we conducted two experiments per dataset, with perturbations chosen as detailed in the Appendix Note 1 (Dataset) section. Would be happy to answer any further details regarding each experiment.
> > > Currently, the MorphoDiff framework does not include additional covariates, such as chemical concentration. However, we recognize that drug dosage can influence cellular phenotypes. Incorporating such informative covariates, like concentrations, could be a valuable enhancement to our framework, potentially improving image generation accuracy. We appreciate this suggestion and view it as a promising direction for future work.

---

### Meta-Review · Area_Chair_X1i6 · 2024-12-23

**Metareview:**

The paper presents a latent image diffusion generative model trained to predict cell morphology conditioned on drug or genetic perturbations, with the goal of predicting the phenotype of novel perturbations in silico. The reviewers strongly recommended acceptance, citing importance of the problem, the high quality execution of the method and validation, and demonstrated biological impact.

**Additional Comments On Reviewer Discussion:**

Given the strength of the original submission, there was little back and forth during the review and revision, with the reviewers largely expressing satisfaction with the paper (including the one reviewer who gave an initial score of 3 and did not update).

---

### Decision · Program_Chairs · 2025-01-22

Accept (Spotlight)